# Hydrological control of river and seawater lithium isotopes

Fei Zhang [1,2 ✉], Mathieu Dellinger [2,3], Robert G. Hilton [2,4], Jimin Yu [5,6], Mark B. Allen [7], Alexander L. Densmore[2], Hui Sun[1] & Zhangdong Jin [1,8,9 ✉]

Seawater lithium isotopes ($\delta^7Li$) record changes over Earth history, including a $\sim 9‰$ increase during the Cenozoic interpreted as reflecting either a change in continental silicate weathering rate or weathering feedback strength, associated with tectonic uplift. However, mechanisms controlling the dissolved $\delta^7Li$ remain debated. Here we report time-series $\delta^7Li$ measurements from Tibetan and Pamir rivers, and combine them with published seasonal data, covering small ($<10^2$ km$^2$) to large rivers ($>10^6$ km$^2$). We find seasonal changes in $\delta^7Li$ across all latitudes: dry seasons consistently have higher $\delta^7Li$ than wet seasons, by $-0.3‰$ to $16.4‰$ (mean $5.0 \pm 2.5‰$). A globally negative correlation between $\delta^7Li$ and annual runoff reflects the hydrological intensity operating in catchments, regulating water residence time and $\delta^7Li$ values. This hydrological control on $\delta^7Li$ is consistent across climate events back to ~445 Ma. We propose that hydrological changes result in shifts in river $\delta^7Li$ and urge reconsideration of its use to examine past weathering intensity and flux, opening a new window to reconstruct hydrological conditions.

[1] SKLLQG, Institute of Earth Environment, Chinese Academy of Sciences, Xi'an 710061, China. [2] Department of Geography, Durham University, Durham DH1 3LE, UK. [3] EDYTEM-CNRS-University Savoie Mont Blanc (USMB), Chambéry 73000, France. [4] Department of Earth Sciences, University of Oxford, Oxford OX1 3AN, UK. [5] Pilot National Laboratory for Marine Science and Technology (Qingdao), Qingdao 266237, China. [6] Research School of Earth Sciences, The Australian National University, Canberra ACT 2601, Australia. [7] Department of Earth Sciences, Durham University, Durham DH1 3LE, UK. [8] Open Studio for Oceanic-Continental Climate and Environment Changes, Pilot National Laboratory for Marine Science and Technology (Qingdao), Qingdao 266237, China. [9] Institute of Global Environmental Change, Xi'an Jiaotong University, Xi'an 710049, China. ✉email: zhangfei@ieecas.cn; zhdjin@ieecas.cn

Silicate weathering influences Earth's climate and habitability by transferring carbon dioxide ($CO_2$) from the atmosphere to the lithosphere[1]. However, understanding how this process has varied in the past and what has controlled it (e.g., through changes in climatic or tectonic forcing) remains a major challenge[2–4]. Lithium isotopes (reported as $\delta^7Li$) have been widely used as a tracer for silicate weathering[5–7]: Li is hosted mainly in silicate minerals[8], whose Li contents are orders of magnitude higher than those of carbonate[8,9] and vegetation[10–13], and its isotopes ($^7Li$ and $^6Li$) fractionate during secondary mineral formation associated with weathering[9,14]. The $\delta^7Li$ value of river water (dissolved Li) is controlled by the congruency of silicate weathering, i.e., the ratio of primary mineral dissolution relative to secondary mineral formation[7,15,16]. Congruent release of Li results in riverine $\delta^7Li$ values that match primary rock compositions with no isotopic fractionation[16–19]. Incongruent release of Li due to uptake of Li into secondary minerals will increase dissolved $\delta^7Li$ values by preferentially incorporating light $^6Li$ into clays[6,16,19–23]. Thus, riverine $\delta^7Li$ does not necessarily directly track silicate weathering rate or intensity (defined as the ratio of silicate weathering to total denudation rate[19,24]).

Past records of the seawater $\delta^7Li$ are characterized by large excursions (by up to 15‰) over $10^6$ year timescales[25] as well as gradual changes over $10^7$ years[25], such as occurred during the Cenozoic when $\delta^7Li$ increased by ~9‰[6]. Different interpretations exist for the Cenozoic $\delta^7Li$ increase[6,24,26,27], but most studies have interpreted it as reflecting a shift from more congruent continental weathering ~60 million years ago (typical of flat lowland settings) to more incongruent weathering in the present-day (typical of mountains), as a result of increased global denudation. This Cenozoic seawater $\delta^7Li$ rise, as with $^{87}Sr/^{86}Sr$ and $^{187}Os/^{188}Os$ changes, was thought to be driven by mid to late Cenozoic tectonic uplift of major mountain ranges[28–31], which would have led to a change of the global weathering rate or the weathering feedback strength[6,24,26,27,32].

A major issue with this interpretative framework is that present-day riverine $\delta^7Li$ data do not follow the expected pattern: some rivers draining flat lowlands have high $\delta^7Li$[9,14], whereas others draining active mountains (e.g., Himalayas, Andes and New Zealand) have generally lower $\delta^7Li$ values relative to downstream areas[7,16,19,33]. Besides, nearly the entire range of riverine $\delta^7Li$ values can be found in fluids within a single weathering profile[34]. These observations, contrary to expectations from shift in weathering regimes, illustrate the difficulty of interpreting modern-day riverine $\delta^7Li$, and complicate our understanding of past $\delta^7Li$ records[7,9,15,16,19,33,35–38]. Weathering in large river floodplains has been proposed as a source of high $\delta^7Li$[7,16,19,35], but the existence and extent of this process is debated[36]. Instead, reactive-transport modelling approaches suggest that fluid residence time may impose a major control on dissolved $\delta^7Li$ and weathering congruency[26,34,38–41], but this idea needs to be confirmed by observations from modern rivers globally.

Here we explore the role of hydrology as a control on present-day global river $\delta^7Li$ variations. We first characterize the seasonal variability in $\delta^7Li$ values by reporting new time-series data from the Tibetan and the Pamir Plateaus. We then revisit published seasonal data of individual rivers from the Arctic to the equator (Supplementary Fig. 1) and compare the spatial average $\delta^7Li$ of global 64 rivers of various sizes and climatic conditions, but from similar geomorphic settings, to assess the influence of hydrology. Finally, we propose a unifying interpretation for past $\delta^7Li$ changes on geological timescales.

## Results and discussion
**Seasonality of global riverine $\delta^7Li$.** The precipitation regime affects the residence time of waters in river catchments, across

storm events and over seasonal timescales[42–44]. Thus, timeseries of riverine $\delta^7Li$ can provide an insight into Li isotopic fractionation as a function of varying duration and/or degree of water-rock interaction[35]. Our weekly-sampled time-series data from the northeastern (NE) Tibetan Plateau (Supplementary Fig. 2) provide a case study for in-depth understanding of the seasonal behaviour of riverine $\delta^7Li$ (Fig. 1a). In the Buha River (BH), riverine $\delta^7Li$ values are the highest (up to 22.4‰) during winter dry conditions, when river water discharge ($Q_w$) is fed by baseflow (minimum $Q_w$ of 1.5 $m^3/s$), corresponding to slow flow and thus a long water-rock contact time. In contrast, at the onset of the summer monsoon, with a sharp increase in $Q_w$ (up to 221 $m^3/s$), BH riverine $\delta^7Li$ decreases to its lowest value of ~12.0‰. Fast river flow during the summer monsoon corresponds to relatively short water-rock interaction times, which is consistent with an increase in the relative contribution of carbonate versus silicate weathering giving rise to a decrease in $^{87}Sr/^{86}Sr$ in this carbonate-dominated catchment[45]. After the summer monsoon, riverine $\delta^7Li$ returns to high values concurrent with decreasing $Q_w$. We observe similar seasonal $\delta^7Li$ variations in the adjacent but silicate-dominated Shaliu River (SL), which shows a seasonal riverine $\delta^7Li$ variation ($\delta^7Li_{dry} - \delta^7Li_{wet}$) of ~8.5‰ (Fig. 1b). Each river shows a negative relationship between $\delta^7Li$ and $Q_w$ (Fig. 1c), and when the weekly data are considered altogether, there is a significant ($r^2 = 0.55$; $P < 0.0001$) negative relationship (Fig. 1c), despite their contrasting lithology (Fig. 1d). The negative correlation is qualitatively supported by our new data from glacial streams in the NE Pamirs (Supplementary Fig. 4): all 10 sampling sites from glacier margins to downstream exhibit systematically higher $\delta^7Li$ values during the dry season (spring, low ice melt) compared to the wet season (summer, high ice melt) (Supplementary Fig. 5a).

We have extended our investigation into other rivers, globally, including both the spatially seasonal dataset (Supplementary Fig. 5) and the time-series $\delta^7Li$ dataset from the very small Strengbach to the large tropical Congo (Supplementary Fig. 6). A common pattern is observed: each river exhibits higher $\delta^7Li$ in dry seasons compared to wet seasons (Fig. 2), with a caveat that a few data show small seasonal $\delta^7Li$ variations less than the analytical error (<±1‰, Supplementary Fig. 7) either due to a low degree of precipitation seasonality (e.g., the Columbia River draining the east of the Cascades[38]) or very similar $Q_w$ of sampling seasons (Supplementary Fig. 8). Overall, the difference in riverine $\delta^7Li$ between dry and wet seasons ranges from −0.3‰ to +16.4‰ (Fig. 2). The Earth's two largest river systems, the Congo and the Amazon Rivers, which together contribute ~20% of the freshwater supply to the oceans, show 7.5‰ differences in $\delta^7Li$ values between dry and wet seasons in the Congo (Fig. 2), and >10‰ at the Amazon mouth[46], respectively. The seasonal data discussed here (6 sets of time-series and 72 seasonal datasets) includes the mainstems and tributaries of several world's largest rivers, i.e., the Amazon, Congo, Ganges, Brahmaputra, Yenisei, Yellow Rivers. The calculated total annual Li flux of these large rivers is $1.44 \times 10^9$ mol/yr (Supplementary Data 1), accounting for ~52% of the Li flux for major world rivers estimated by Huh et al.[14]. As with any river sample set, spatial gaps in continental coverage are inevitable. However, the rivers investigated here are globally representative: they form a dataset that covers a large range of vastly contrasting climates and vegetation (from high-latitudes to the equator), basin sizes (from small catchments to Earth's largest rivers, Supplementary Fig. 9), and geomorphic settings (Arctic permafrost, Rocky and Andean mountains, Loess Plateau, Pamir-Tibetan Plateaus, Himalayan floodplains, and tropical rainforests). Such consistent seasonal $\delta^7Li$ pattern differs completely from the temporal variability of Sr isotopes in global rivers, i.e., all seasonal and time-series $\delta^7Li$ across latitudes

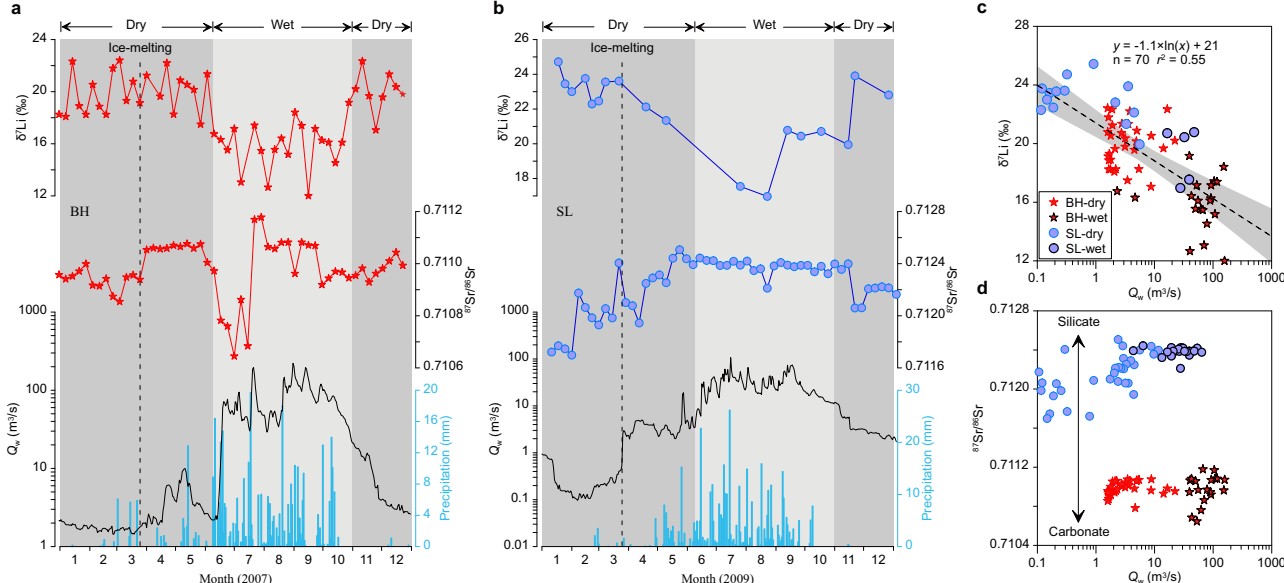

**Fig. 1 High-resolution river water $\delta^7$Li, $^{87}$Sr/$^{86}$Sr, and hydrometeorological data from the NE Tibetan Plateau.** Weekly variations of $\delta^7$Li and $^{87}$Sr/$^{86}$Sr in the carbonate-dominated BH (**a**) and silicate-dominated SL (**b**) catchments (Supplementary Fig. 2) along with daily $Q_w$ and precipitation, showing inverse trends between $\delta^7$Li and $Q_w$ in each river. When plotting up weekly data from the two rivers together (**c**), there is still an overall negative relationship, highlighting a strong hydrology control on riverine $\delta^7$Li. (**d**) $^{87}$Sr/$^{86}$Sr versus $Q_w$, showing large differences between the two rivers, reflecting their distinct lithology (Supplementary Fig. 3). The dashed lines in **a** and **b** represent ice-melting times. Errors for $\delta^7$Li are <0.9‰. The shaded regions in **c** show 95% confidence intervals. Symbols with black borders in **c** and **d** represent wet seasons, and others are dry seasons.

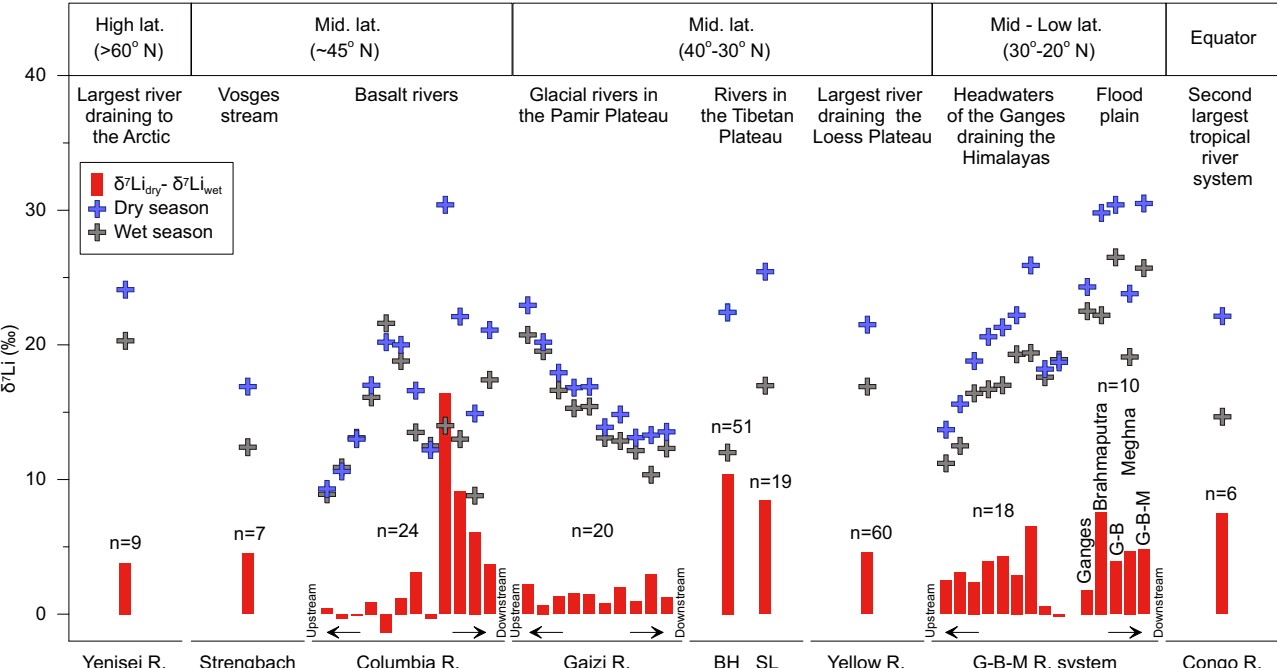

**Fig. 2 Seasonal differences in river water $\delta^7$Li across latitudes.** Mainstreams and tributaries in river basins from ~1 to >10$^6$ km$^2$ in drainage area show systematically higher $\delta^7$Li values in dry seasons ($\delta^7$Li$_{dry}$, blue crosses) than those in wet seasons ($\delta^7$Li$_{wet}$, grey crosses). For time-series data (weekly or monthly samples at one sampling site) in the BH and SL (Fig. 1, this study), Yenisei[48], Strengbach[10], Yellow[47], and Congo[37] (Supplementary Fig. 6), the highest values in dry seasons and lowest in wet seasons are presented to explore their maximum differences (Red bars, defined as: $\delta^7$Li$_{dry} - \delta^7$Li$_{wet}$). For other seasonal data, the Columbia[38], Gaizi (this study), and Ganges-Brahmaputra-Meghna River (G-B-M R.) systems[8,39] show spatial spot samples from upstream to downstream in each basin, with samples collected both in dry and wet seasons for each sample site (Supplementary Fig. 5). Red bars represent their seasonal differences by sample sites. In the Gaizi River, seasonal data are distributed from glacier margins to downstream. In the G-B-M R. system, seasonal data are presented from the small headwater of the Ganges[8] to large main tributaries (Ganges, Brahmaputra, Meghna) and further to downstream mainstem (G-B-M)[39]. "n" is the number of the data collected for each basin. See Supplementary Note 1 and Data 1 for additional details and data sources. Errors for $\delta^7$Li are similar to the symbol size.

investigated here show lower values in wet relative to dry seasons. By contrast, time-series $^{87}Sr/^{86}Sr$ values display both increasing and decreasing trends from dry to wet seasons (Supplementary Fig. 10 and Data 3). The difference between Li and Sr isotopes suggests distinct control mechanisms, with the $^{87}Sr/^{86}Sr$ probably reflecting lithological variability as shown by Sr data from the BH and SL catchments (Fig. 1d; Supplementary Fig. 3).

Previous interpretations suggest that seasonal $\delta^7Li$ variation in individual rivers could reflect tributary mixing[37], seasonal temperature shifts[47], or influence of fluid residence times[38,39,48]. If mixing of different water bodies with distinct $\delta^7Li$ is the main control, $\delta^7Li$ shifts from dry to wet seasons should either increase or decrease, depending on isotopic compositions and $Q_w$ of tributaries. The consistency of $\delta^7Li$ across rivers of vastly different catchment areas and network structures indicates that tributary mixing cannot explain all observations (Supplementary Note 1). A significant temperature control can also be excluded for some river basins characterized by the relatively small seasonal air temperature variation, in particular in the Congo (Supplementary Fig. 11) and Amazon[46] rainforests, and no correlation between temperature and $\delta^7Li$ in the Yangtze River headwaters[33]. In addition, groundwater contribution cannot explain the consistently elevated $\delta^7Li$ of river waters during dry seasons (Figs. 1 and 2). Because groundwaters have both low and high $\delta^7Li$ values[49], ranging from +6‰ to +29‰, and thus can either raise or lower river water $\delta^7Li$. Moreover, several rivers show higher $\delta^7Li$ than that of groundwaters[13,34,35,38,39]. Direct input of groundwaters with lower $\delta^7Li$ (e.g., in the Ganges-Brahmaputra basins[39]) would decrease river $\delta^7Li$, which is at odds with the consistent increase in $\delta^7Li$ in dry seasons across latitudes (Figs. 1 and 2). By the same token, the observed consistent decreases in river $\delta^7Li$ with increasing $Q_w$ during wet seasons argue against a first order control of groundwaters.

While human activities may significantly increase river Li level and decrease $\delta^7Li$ in the Han River[50], our assessment suggests that anthropogenic activities do not seem to have a widespread impact on large river basins with high population densities, although anthropogenic Li influences deserve further attention (see details in Supplementary Note 3). A recent study[51] in relatively dry Loess Plateau proposes that evaporation can increase river water $\delta^7Li$ via enhanced secondary mineral precipitation in soil waters. However, within the BH and SL catchments, stronger evaporation occurs during wet seasons (Supplementary Fig. 12), but river waters bear lower $\delta^7Li$ values, with a negative relationship between $\delta^7Li$ and $Q_w$ (Fig. 1c). We also noted that in some humid regions such as Congo tropical rainforest that are characterized by roughly stable temperature and evaporation, riverine $\delta^7Li$ variations are very sensitive to seasonal $Q_w$ variations (Supplementary Fig. 11b). Overall, we believe evaporation plays an insignificant role in affecting riverine $\delta^7Li$.

Here we argue that the water residence time control on Li isotopes can be generalized globally to explain the observed seasonal variability. As dry and wet season river water tend to have long and short water residence times, respectively[42,48], we suggest that higher $\delta^7Li$ in dry seasons can be attributed to a larger amount of secondary mineral formation from fluids with longer water-rock interactions (which could promote mineral saturation and thus formation; Supplementary Fig. 13), leading to a higher proportion of $^6Li$ incorporated in clays and higher $\delta^7Li$ values in river waters. The role of residence time is strongly supported by laboratory experiments where increases in water-rock interaction time resulted in continuous and large (12‰) increases in $\delta^7Li$ in solutions in 12 days, and more than 16‰ in a month[52]. Furthermore, substantial increases (~8‰) in solution $\delta^7Li$ driven by longer residence time is confirmed by dissolution

experiments with loess samples over 10 days (Supplementary Fig. 14). These experiments indicate that fluid residence time can effectively cause large solution $\delta^7Li$ changes over time scales that are comparable to seasonal variations observed in natural river systems. Additionally, cave drip water data also support an important role of fluid residence time in $\delta^7Li$ variations on monthly to seasonal timescales[40].

**Spatial river $\delta^7Li$ and annual runoff.** The hydrological control on riverine $\delta^7Li$ is not only observed in seasonal variations, but also when comparing different river systems. We compiled all published riverine $\delta^7Li$ data from specific geomorphic settings (i.e., rivers draining lowlands or mountains) and for which runoff data was available, to isolate the influence of hydrology (see Methods). Our compiled $\delta^7Li$ from medium to large rivers ($10^3$ km$^2$ to >$10^6$ km$^2$ in size) draining only flat lowlands (i.e. having similar geomorphic setting) show different $\delta^7Li$ values: dry, middle-to-high latitude rivers have higher values than wet, tropical rivers (Fig. 3a). The negative relationship between the seasonally averaged $\delta^7Li$ and annual runoff for lowland rivers (Fig. 3b) can be interpreted in the same way as seasonal hydrological shifts in each river: cold, drier conditions with lower precipitation lead to a longer fluid residence time, improved or greater mineral saturation, more secondary mineral formation, and thereby higher riverine $\delta^7Li$ values.

In comparison, rivers draining only mountain ranges, characterized by high rates of weathering and erosion (Supplementary Fig. 15), exhibit low $\delta^7Li$, and importantly, data from these settings plot on a similar trend defined by lowland rivers with the exception of some New Zealand rivers (Fig. 3b). One plausible explanation for the similar $\delta^7Li$ between mountain rivers and tropical lowland rivers, despite having very different topography and erosion rates (Supplementary Fig. 15), could be their similarly higher runoff (Fig. 3b) and therefore shorter water residence times than dry, middle-to-high latitude rivers. We note that the river basins compiled here differ largely in other aspects such as sediment concentration, mineralogy, weathering rate/intensity, vegetation, and geomorphic settings, which all have been proposed to affect Li isotopic fractionation[7,9,12,14,16,19,26,33,36,38], yet there is still a consistent hydrological control, suggesting that a common mechanism governs temporal and spatial $\delta^7Li$ on the continents. The relatively small deviation of $\delta^7Li$ within each geomorphic setting imply that superimposed upon the major and common hydrology imprint, other factors (e.g., topography, vegetation, soil thickness) may, to some extent, also contribute to the riverine $\delta^7Li$ variability (Supplementary Fig. 16).

**$\delta^7Li$ evolution over geological timescales.** In addition to the modern river dataset (Figs. 2 and 3), there is evidence for an important hydrological control on $\delta^7Li$ from geological archives across a range of timescales. First, speleothems from two Israeli caves[53] record high $\delta^7Li$ values (~23‰) during drier, glacial periods, and low values (~10‰) during wetter inter-glacials (Fig. 4c; Supplementary Fig. 17). Second, over million-year timescales, several climatic events are characterized by changes of seawater $\delta^7Li$ during the Cenozoic. The lowest seawater $\delta^7Li$ value (~22‰) is recorded during the Paleocene-Eocene Thermal Maximum (PETM)[6] (Fig. 4d), with rapid ~3‰ negative excursion of $\delta^7Li$ over only ~100 kyr[54], when precipitation and continental runoff was dramatically strengthened in a much warmer world than today[54–58]. Similarly, the Early Eocene Climatic Optimum (EECO) had much lower seawater $\delta^7Li$ (~23‰) than the present-day (Fig. 4d). Proxy records suggest the period was broadly coincident with a shift to wetter climate[56,59], characterized by enhanced erosion and weathering compared to the non-

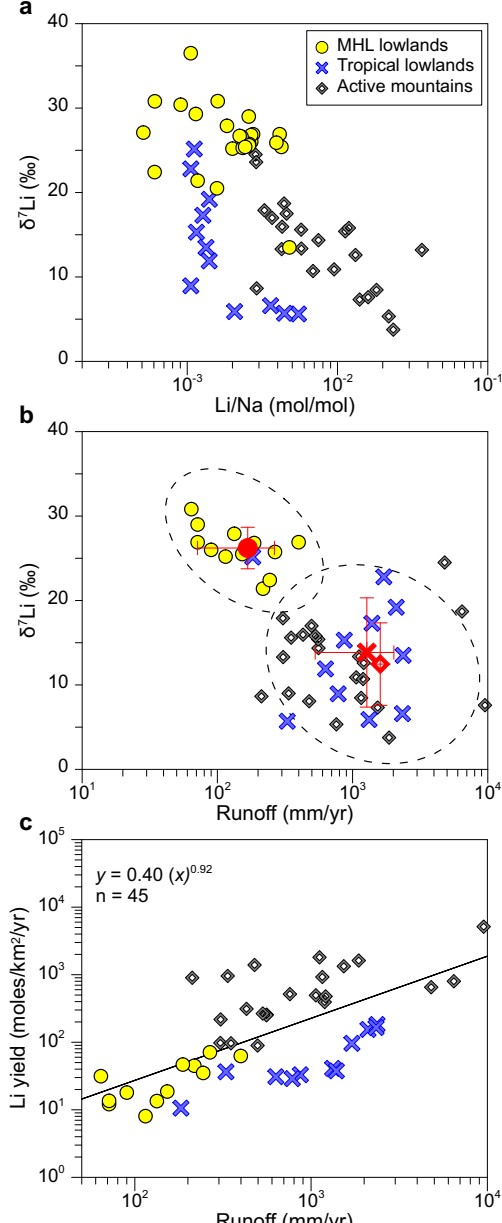

**Fig. 3 Riverine $\delta^7$Li and Li yield from various geological settings.** (**a**) $\delta^7$Li versus Li/Na. Tropical lowlands (blue crosses) and mountain areas (grey squares) show similar $\delta^7$Li values, but have lower $\delta^7$Li than middle-to-high latitude (MHL) lowlands (yellow dots). (**b**) $\delta^7$Li versus runoff, showing an overall negative correlation between MHL and tropical lowlands. Red symbols are averages of $\delta^7$Li and runoff in the MHL (red dot), tropical lowlands (red cross) and mountain areas (red square). Red errors are standard deviations. Two dashed ellipses cover the majority of the MHL and tropical lowlands, respectively. (**c**) Dissolved Li yield versus runoff, showing higher runoff with higher Li yield. There are fewer data points of MHL lowlands in **b** and **c** because no runoff data is available for some rivers. The MHL lowland dataset includes 27 rivers draining the Greenland Shield, Canadian Shield, Siberian Shield, and Baikal Rivers. The tropical lowland includes 12 rivers draining the Amazon Shield, Orinoco Shield, and the Congo River. The active mountain dataset includes 25 rivers draining the Andes, New Zealand Alps, Himalaya, Rocky and the Mackenzie Mountains, and upstream of the Yangtze, Mekong, and Salween Rivers (see methods).

glacial Quaternary[60]. The Mid Miocene Climatic Optimum (MMCO) also shows a negative Li isotopic excursion[6]. Evidence from plant leaf wax $\delta$D and detrital sediment records indicate an intensified hydrological cycle in the Antarctic and the NE Tibetan Plateau at this time[61,62]. Third, extending to older times beyond the Cenozoic, the Cretaceous Ocean Anoxic Events OAE1a and OAE2, characterized by rapid increase in $p$CO$_2$ and global warming, show abrupt declines of marine carbonate $\delta^7$Li (Fig. 4e, f), consistent with accelerated hydrological cycles[5,63–66].

The Hirnantian glaciation (~445 Ma) recorded a global temperature drop of 8–10 °C, culminating in an ice-sheet over Gondwana and subsequent global sea-level fall[67,68]. Marine carbonate $\delta^7$Li values show a positive excursion during this event (Fig. 4g). Sedimentary evidence indicate a climate shift from warm, humid to overall cold, arid[67,69], consistent with the positive $\delta^7$Li excursion suggesting a 4-fold reduction in global weathering flux[68].

To understand the past variability in the $\delta^7$Li of seawater, it is also important to consider any related changes in the dissolved Li flux to the ocean (Fig. 3 and Supplementary Fig. 18), as this can impact the residence time of Li in the ocean (Supplementary Note 2). The above-mentioned geological events support that the modern/Neogene/OAEs Li ocean mass balance were probably fundamentally different and residence times were much shorter in the past[5,63,65]. This was implied by modeling the OAE2 event (i.e., an increase in river Li flux results in a decrease in seawater Li residence time and $\delta^7$Li value[5]), and is also indicated at the timescales shown in Fig. 4. Together, all the events described herein display a response of $\delta^7$Li to climate change that is consistent with present-day riverine observations, i.e., $\delta^7$Li values decrease when climate becomes wetter which ensues shorter average fluid residence time in the continental weathering zone, and vice versa.

**The role of hydrology in Cenozoic seawater $\delta^7$Li evolution.** We have shown that hydrology exerts a primary control on riverine $\delta^7$Li values over seasonal and annual timescales, across latitudes and basin sizes (Figs. 2 and 3). These patterns can be explained by changes in the mean fluid residence time in a river basin, which influences the degree of Li isotope fractionation between primary minerals and the fluid phase via secondary mineral formation. The results from modern rivers are broadly coherent with Li isotope ratio shifts across pronounced changes in climate over tens of thousands (glacial cycles) to millions of years (e.g., PETM, OAEs) (Supplementary Fig. 17; Fig. 5). The associated large and rapid $\delta^7$Li shifts (e.g., ~13‰ in OAE2 with a duration of only ~440 kyr[5]) appear too rapid to be linked to tectonic processes, but instead are consistent with a common hydrological change (Fig. 4e–g). Altogether, we propose that climate-driven hydrological changes alone can produce large $\delta^7$Li shifts over various timescales investigated here.

These findings may map onto the long-term Cenozoic shift in $\delta^7$Li values of seawater. Previous studies have proposed that increasing Cenozoic seawater $\delta^7$Li, $^{87}$Sr/$^{86}$Sr and $^{187}$Os/$^{188}$Os values towards the present could reflect increases in global weathering fluxes linked to mountain uplift, but this remains intensively debated[2–4,16,32]. More recently, marine beryllium (Be) isotope records and marine calcification indicators were interpreted as reflecting either increased[70], or near constant[2], or a decreased[3] continental weathering flux during cooling of the Neogene, or even the Cenozoic[4] (Fig. 5). Combined with $^{10}$Be/$^9$Be, new interpretation of seawater $\delta^7$Li records infers an increase in the feedback strength of silicate weathering (or increase in land surface reactivity) along with a stable weathering flux driven by rock uplift[32]. This means that even though global climate cooled, weathering fluxes did not decline, because silicate weathering became more sensitive to climate over this period. Model results

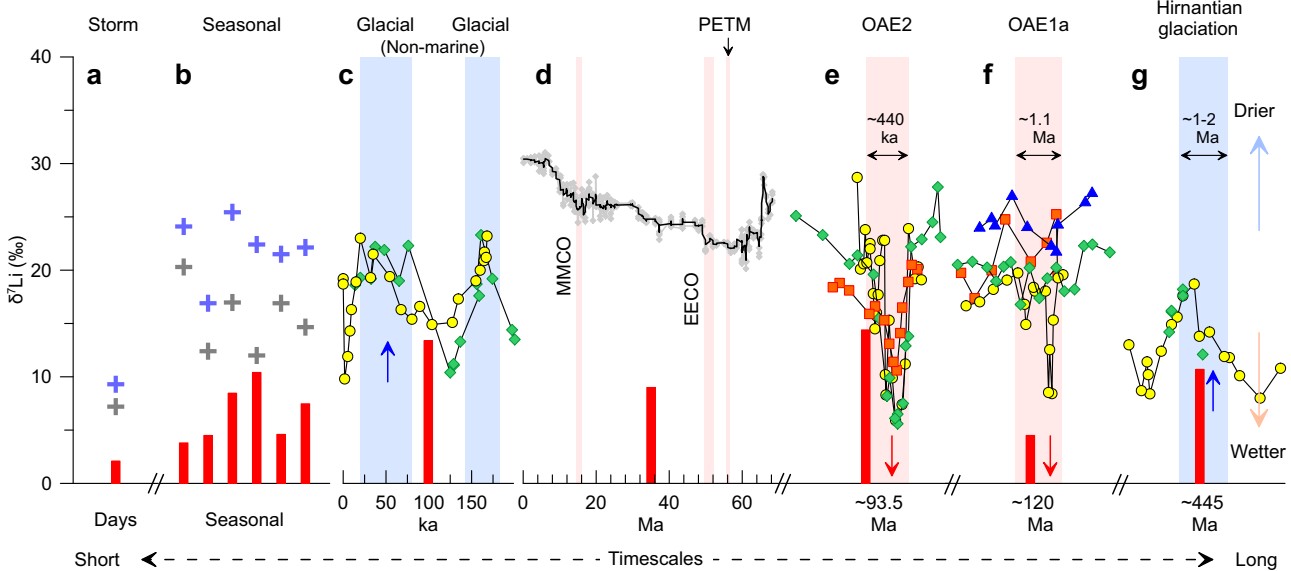

**Fig. 4 Temporal evolution of $\delta^7$Li on various timescales ranging from days to months, millennial, and million years.** (**a**) Storm events in tropical Guadeloupe showing a decrease of stream $\delta^7$Li from 9.3‰ to 7.8‰ within 1–2 days[92], similar to the onset of monsoon at Tibetan rivers (Fig. 1). (**b**) Seasonal (time-series) variations of rivers from the Arctic to the equator, showing systematically elevated $\delta^7$Li in dry seasons. (**c**) Speleothems $\delta^7$Li from two Israeli caves during the last glacial cycle[53]. (**d**) Seawater $\delta^7$Li evolution during the Cenozoic[6]. Pink bars mark global climate events, including Paleocene-Eocene Thermal Maximum (PETM, ~56 Ma), Early Eocene Climatic Optimum (EECO, 50–52 Ma) and Mid Miocene Climatic Optimum (MMCO, 14–17 Ma). (**e**) and (**f**) Marine carbonate $\delta^7$Li of two major Ocean Anoxic Events (OAE1a, ~120 Ma)[63] and (OAE2, ~93.5Ma)[5]. (**g**) Marine carbonate $\delta^7$Li during the end-Ordovician Hirnantian glaciation (~445Ma)[68]. The events of OAE2, OAE1a and Hirnantian glaciation lasted for ~440ka[5], ~1.1Ma[63] and ~1–2Ma[68], respectively. Blue bars for dry events in **c** and **g** exhibit positive $\delta^7$Li excursions. Pink bars for wet events in **d**, **e** and **f** show lower $\delta^7$Li values (EECO and PETM), and negative $\delta^7$Li excursions (MMCO, OAE2 and 1a). The red bars in **c**–**g** show the excursion amplitudes of $\delta^7$Li (defined as $\delta^7$Li$_{dry}$ - $\delta^7$Li$_{wet}$) during the events, with blue and red short arrows showing positive and negative excursion, respectively. See Fig. 2 for the legends in **a** and **b**. Different symbols in **e** and **f** represent marine carbonate sections at different locations, and in **g** represents bulk carbonates and brachiopods at same location. Errors for $\delta^7$Li are similar to the symbol size.

suggest this came about because of increased erosion in tectonically active mountains, making more of the terrestrial land surface locations where silicate weathering was no longer limited by supply of minerals, and instead primarily controlled by runoff and temperature[4,32,71,72].

Based on the data herein (Figs. 1–4), we propose to add a complementary piece to the puzzle of Cenozoic evolution: overall drying of climate during cooling since 50 Ma has left an imprint on marine $\delta^7$Li records (Fig. 5). A number of explanations[6,24,26,27] for the 9‰ positive shift in seawater $\delta^7$Li over the last ~50 Ma have converged on the need for a significant increase in the riverine $\delta^7$Li values[27]. This could be achieved by a less intense continental hydroclimate that resulted in higher $\delta^7$Li of continental runoff (Fig. 3 and Supplementary Fig. 18). Any associated Li flux reduction would increase the Li residence time in the ocean and thereby increase seawater $\delta^7$Li (Supplementary Note 2), similar to inferences from global climatic events, e.g., the rapid decline of $\delta^7$Li during OAE2 that suggests a 2–4 times increase in river fluxes and 25%–50% decrease in seawater Li residence times relative to the present day, coupled with wetter continental conditions[5].

The proposed drying climate over the Cenozoic is consistent with a >120 m drop of sea level[73] and step-wise aridification recorded in many regions[74–77] (e.g., central Asia, North America, Europe, Africa, and Australia). In addition to million-year records[76], modern meteorological data confirm the positive relationship between global-mean temperature and precipitation[78–80], with a 4% decrease in global runoff per °C cooling[78]. Given a cooling Cenozoic, these observations suggest lower continental runoff. Critically, our proposed hydrological control on the Cenozoic seawater $\delta^7$Li evolution is strongly supported by a clear negative trend between $\delta^7$Li and continental mean annual precipitation

from the Pacific and Atlantic sides of Eurasia during the last 50 Ma (Fig. 5d–e).

This proposed mechanism, the reduction of continental runoff (and increased time for water-rock interaction) controlling long-term seawater $\delta^7$Li, raises questions on how silicate mineral weathering linked to hydrology mediates global climate over the Cenozoic. There are two possibilities: (i) that this led to decreased weathering fluxes; (ii) it led to no change in weathering rate if compensated by an increase in the strength of the silicate-weathering climate feedback[32] and/or by changes in $CO_2$ release from volcanism[81,82], metamorphism in continental arcs[83] or sedimentary rock weathering[31,84–86]. In terms of the first scenario, based on our dataset from flat lowlands and active mountains, a lower runoff would result in a lower silicate weathering rate (Supplementary Fig. 15c). This would also be consistent with other observations[71,72] that suggest a 1% decrease in global runoff is accompanied by a 0.4–0.7% decrease in solute fluxes, while being in line with new data from the Himalayan-Tibetan areas that show no increase in weathering[87] and erosion[88] over these timescales. Together, this would challenge the long-standing uplift/weathering hypothesis[28].

Recent studies have suggested global stability of the chemical weathering flux during the Cenozoic[2], which supports the second scenario. This could come about if declining atmospheric $CO_2$ was primarily driven by decreasing solid earth degassing rates, as proposed by recent studies[81–83], which would require stable silicate weathering rates while atmospheric $CO_2$ declined. The release of $CO_2$ through weathering of sedimentary rocks[31,84–86] could also play a role in net carbon cycle balance. Finally, a change in the strength of the feedback between climate and silicate weathering rate could have occurred[32,72,89]. A

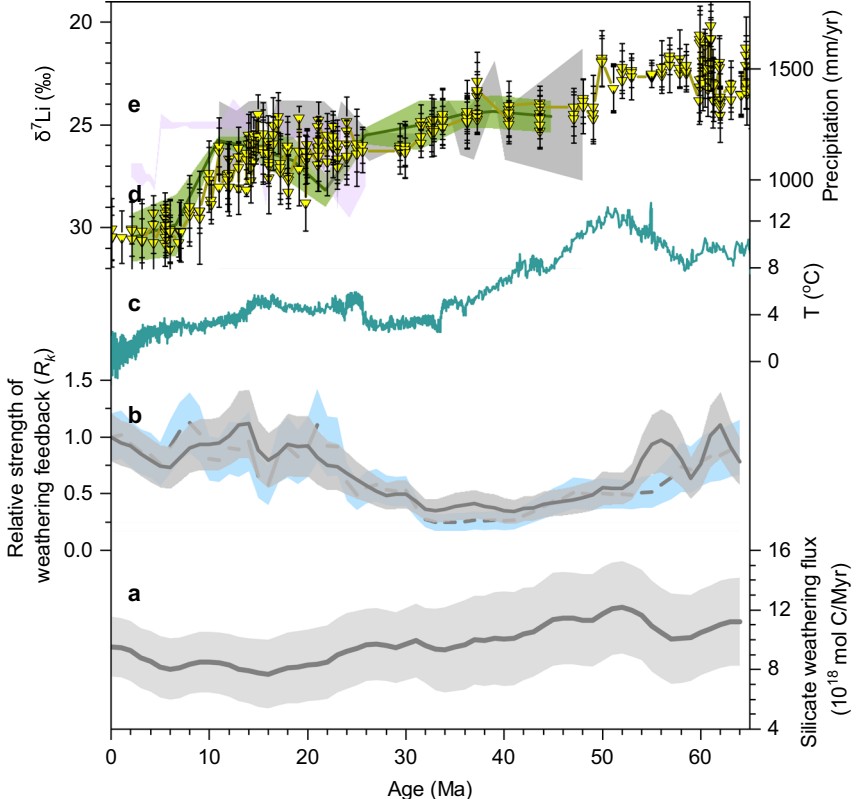

**Fig. 5 Geological δ⁷Li records, climate and weathering flux.** (**a**) Silicate weathering flux (grey solid line)[4]. (**b**) Relative strength of weathering feedback, with dark and light blue shadings indicating low and high atmospheric $CO_2$ scenarios[4], respectively. (**c**) Deep ocean temperature (dark green)[73]. (**d**) Seawater δ⁷Li (yellow) evolution during the Cenozoic[6]. (**e**) Continental mean annual precipitation from the Pacific (green) and Atlantic (grey and light purple) sides of Eurasia[75]. During the Cenozoic cooling, increasing δ⁷Li coincides with decreasing precipitation recorded at the two sides of Eurasia (**e**) and are likely coupled with a stable or decreased silicate weathering rate (**a**). Shadings in **a** and **b** indicate 1σ uncertainty, and in **e** show mean annual range of precipitation.

strengthening of the weathering-climate feedback[32] should be expected due to increased land surface reactivity related to mountain building and higher physical erosion[28,90]. Our dataset and this weathering-feedback scenario would compensate each other, i.e., the Cenozoic δ⁷Li increase is driven by the lengthening of the average residence time of water on the continents as the climate cooled and runoff declined. This weakened hydrology and weathering (Supplementary Fig. 15c), combined with increased feedback strength caused by late Cenozoic uplift (Fig. 5b), could have sustained weathering fluxes even as cooling proceeded[91]. The extent to which this increased feedback strength can offset the decreased weathering flux caused by weakened hydrology is beyond the scope of our current study and needs more quantitative constraint from future research.

In summary, there are multiple lines of evidence across a range of temporal and spatial scales that support a strong control of continental hydrology (via fluid residence time) on Li isotopes. By recognizing a hydrological control on the δ⁷Li of continental runoff, our findings call for a renewed focus on how changing hydrological regimes affect Earth's weathering and the carbon cycle over tens of millions of years, and during more rapid changes in warming global climate. The link between hydrology and δ⁷Li, from seasonal to geological timescales, makes it as a useful tool to reconstruct past hydrological changes, for which long-term, continuous terrestrial records remain extremely limited[58,60].

## Methods

**Hydrological data**. The daily river water discharge ($Q_w$) of the Buha (BH) and Shaliu (SL) Rivers from 2007 to 2009 were monitored at the Buha and Gangcha hydrological stations, respectively (Supplementary Fig. 2).

**Sample collection**. A total of 103 river water samples were collected weekly from the BH and SL Rivers at the Buha and Gangcha hydrological stations in 2007 and 2009, respectively (Supplementary Fig. 2). Twenty river water samples were collected at 10 sampling sites from glacial margins to downstream during two field campaigns in summer 2014 and spring 2016 at the Gaizi River, NE Pamir Plateau (Supplementary Fig. 4). All water samples were filtered on site through 0.2 μm Whatman® nylon filters. The samples were collected into a 60 mL polyethylene bottle pre-acidified with 6 M quartz-distilled $HNO_3$ and acidified to pH < 2. All samples were kept chilled until analysis.

**Analysis**. Lithium concentrations of all river water samples were analyzed by PerkinElmer NexION 300D ICP-MS at the State Key Laboratory of Loess and Quaternary Geology (SKLLQG) with rhodium as an internal standard. The analytical precision is better than 5%. A total of 90 samples were selected for the measurements of lithium isotope ratios. Detailed pretreatment and measurement procedures were conducted following Gou et al.[47]. Each water sample containing 300 ng Li was dried and purified by single-step cation exchange chromatography filled up with 8 mL resin (Bio-rad® AG50W X-12, 100–200 mesh), with 0.5 M $HNO_3$ as an eluent. Analyses were performed on a Thermo Neptune plus multi-collector inductively coupled plasma mass spectrometer (MC-ICP-MS) at the SKLLQG. A seawater reference material (NASS-6) was analysed as an unknown and repeated measurement over a one-year period yielded a δ⁷Li value of +31.1 ± 0.7‰ (2σ, n = 15), in agreement with the global average seawater value of +31.0 ± 0.5‰ (ref. [6]). Our long-term external reproducibility is better than ± 0.9‰ (2σ) for δ⁷Li measurements[47]. Li isotopes are all reported relative to the standard L-SVEC. For 87Sr/86Sr analysis, the pretreatment and measurement procedures were conducted following Jin et al.[45]. Each water sample containing 100 ng Sr was dried and purified by Eichrom Sr^SPEC exchange column (0.5 mL bed volume each column) preconditioned with 3 M $HNO_3$ and eluted with 4 mL UHQ (ultra high quality) deionised water. All 53 weekly SL River water samples were measured on a MC-ICP-MS in the Isotope Geochemistry lab at the Taiwan Cheng Kung University. Standard reference material NBS 987 (recommended value = 0.710245) was periodically measured to check accuracy. Replicate analyses of NBS 987 yielded an average 87Sr/86Sr ratio of 0.710255 ± 0.000022 (2σ, n = 42). The reported uncertainties were much less than the large 87Sr/86Sr ranges of weekly samples from 0.711700 to 0.712500. Raw 87Sr/86Sr ratios for all samples and standards were corrected for mass bias by normalizing to 86Sr/88Sr = 0.1194 and

corrected for [87]Rb and [86]Kr isobaric interferences. The blank Sr (<1 ng) was less than 1% of the processed water samples. Results for Li and Sr isotope analyses were compiled in Supplementary Data 1.

**Spatial comparison of lithium isotopes at lowlands and mountain rivers**. We compiled a set of river data corresponding to rivers draining either only active mountain ranges or flat lowland areas (Supplementary Data 2). The goal of this compilation is to compare the largest river catchments (because they integrate large areas) having: (1) similar geomorphic settings (flat lowland shields) but different climatic conditions from high latitudes to the equator; (2) contrasting geomorphology between flat lowland and active mountain ranges. Herein, we used the most downstream sample data corresponding to the same geomorphic setting. For mountain ranges, this strictly corresponds to rivers that were sampled upstream floodplain areas since it has been suggested that additional weathering reactions may take place in floodplains[7,19]. Our compilation includes 25 rivers from major orogenic belts (the Andes, the Himalayas-Tibetan Plateau, the New Zealand Alps, the Rocky and Mackenzie Mountains, and upstream of the Yangtze, Mekong, and Salween Rivers) for which runoff data were available. The sediment fluxes for these rivers are higher than 100 t km$^{-2}$ yr$^{-1}$ (high erosion rate) and the runoff range between 212 and 9526 mm yr$^{-1}$.

For lowland rivers, our compilation includes 12 rivers in the tropical lowlands across the equator (the Amazon, Congo and Orinoco Rivers) and 27 rivers in the lowlands at middle-to-high latitudes (the Mackenzie, Lena, Yenisei, and Baikal Rivers). The middle-to-high latitudes and tropical lowland rivers are chosen from rivers draining similar relatively flat topography with low erosion rates (erosion rate < 100 t km$^{-2}$ yr$^{-1}$) to isolate the influence of climate. For each river, we used a single or an average of all the δ[7]Li measurements at the same sampling and the annual runoff. For some rivers, the runoff and sediment flux are not available and therefore not shown on Fig. 3. However, the difference in the average and range of δ[7]Li is minor between rivers with and without runoff data.

**Lithium isotopes of geological records**. We further compiled extensive δ[7]Li data from events across the last glacial cycle to end-Ordovician Hirnantian glaciation (~445 Ma). The dataset includes δ[7]Li of speleothem records from two Israeli caves during the last glacial cycle (from the present to 200 ka)[53], δ[7]Li of foraminifera samples from 8 ocean drill sites during the Cenozoic (60 Ma)[6], δ[7]Li of 3 marine carbonate sections at the Eastbourne and South Ferriby, UK, and Raia del Pedale, southern Italy recording the Ocean Anoxic Event 2 (OAE2, ~93.5 Ma)[5], δ[7]Li of 4 marine carbonate sections recording the Early Aptian OAE1a (~120 Ma)[63], and the δ[7]Li of bulk carbonates and brachiopods from Anticosti Island, Canada (Pointe Laframboise) recording the Late Ordovician Hirnantian glaciation (~445 Ma)[68]. Carbonate-based Li isotope records spanning the past 3 billion years indicate that bulk carbonates can be used to faithfully reconstruct seawater Li isotope values in the deeper past[25].

## Data availability

The authors declare that all data supporting the results of this study are available in the Supplementary Information and Supplementary Data 1 to 3.

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

## Acknowledgements

Conversations with J. Gaillardet regarding seasonal data of the Tibetan Plateau, and M. Murphy for Li isotopes at Durham University are greatly appreciated. We sincerely thank F. von Blanckenburg for providing the data of Mediterranean beryllium isotopes over glacial cycles, and J. Caves Rugenstein for providing the data of weathering feedback strength over the Cenozoic. We are very thankful to L. Deng, Y. Xie, Y. Peng, and Y. Hu

for help in sample collection and laboratory assistance. This work was supported by the NSFC (41991322, 41930864) to Z.J., the Strategic Priority Research Program of Chinese Academy of Sciences Grants (XDB40020502), the Youth Innovation Promotion Association of CAS (E029070299), NSFC (41403111) and full funding to visit Durham University from the China Scholarship Council to F. Z.

## Author contributions

F.Z. conceived the idea, compiled global seasonal and geological data and led the study with M.D., M.D. contributed datasets of spatial lowland and mountain rivers; Z.J. and F.Z. collected new samples in the Tibetan and Pamir Plateaus and performed laboratory measurements; F.Z., M.D., R.G.H., M.B.A, J.Y. and A. D. drafted the manuscript; H.S. provided global precipitation data in the Supplementary Fig. 1; F.Z. and Z. J. managed the project; All authors contributed to the discussion and interpretation of the data and provided valuable input to the manuscript.

## Competing interests

The authors declare no competing interests.
