## [Peer Review File · Nature Communications]

Hydrological control of river and seawater lithium isotopesREVIEWER COMMENTS

Reviewer #1 (Remarks to the Author):

Review of Hydrological control of river and seawater lithium isotopes by Zhang et al.

It was a pleasure reading the manuscript by Zhang et al. In my opinion it's one of the most lucid manuscripts I have read in a while. The work presented here utilizes published data in association with new time-series data to provide new insights into what controls the lithium isotopic composition of river water. Lithium concentration and isotopic composition of river water has been widely applied to quantify the rate of terrestrial silicate weathering and thus, the rate of CO₂ drawdown by weathering. This work proposes a new explanation for the long-term (Cenozoic) changes in seawater lithium isotopic composition to better constrain the role of silicate weathering in drawing down atmospheric CO₂ concentration over this period to usher in modern climate. I would recommend this manuscript for publication with a few key changes / additions.

The major comments are the following:

Firstly, when comparing historical $\delta^{7}\text{Li}$ record of seawater it is risky to include data from studies that utilized bulk carbonates. We now know that different carbonate archives have a different absolute $\delta^{7}\text{Li}$ value when grown in the same seawater matrix (probably vital effect). Seawater $\delta^{7}\text{Li}$ excursions beyond the foraminifera archive is reconstructed from bulk carbonates and focuses on events of abrupt climate change. My strong recommendation is to exclude / minimize the discussion on parts of the paleo seawater $\delta^{7}\text{Li}$ record based on bulk carbonates.

Secondly, the impact of a variable seawater Li residence time on the interpretation drawn by the authors should be incorporated in the manuscript. We are dealing with a classical scenario of changes in river flux and composition vs. seawater $\delta^{7}\text{Li}$ composition. The rate of change of $\delta^{7}\text{Li}$ of seawater is expected to be a direct function of Li residence time. When dealing with a long-term record sticking to modern day residence time estimates should be avoided. I would recommend that the authors include a variable Li residence time in their discussion.

Thirdly, the impact of clay cannibalization (as demonstrated on Rio Negro samples by co-author Dellinger) should be discussed in detail in the manuscript. This process has the ability to strongly impact the $\delta^{7}\text{Li}$ of tropical rivers during low flow (characterized by long residence time of water). Dellinger established that the impact of this process will influence the [Li] and $\delta^{7}\text{Li}$ of the dissolved phase in a way identical to that of congruent weathering of rocks. I surprised that this aspect of Li fluvial chemistry was completely omitted from the manuscript.

There is no mention of possible ground water contribution in controlling $\delta^{7}\text{Li}$ of rivers during low flow – this must be touched upon. There is a limited data on $\delta^{7}\text{Li}$ of ground water; however, recent work from Josh West's group have put realistic constrains on the $\delta^{7}\text{Li}$ of groundwater in Bengal basin. I would recommend the authors utilize that study to discuss possible influence of groundwater on riverine Li chemistry.

Cheers!
Sambuddha

Reviewer #2 (Remarks to the Author):

Review of Zhang et al. "Hydrological control of river and seawater lithium isotopes"

Summary

Zhang and coauthors provide new high-temporal resolution river Li isotope data from Tibet and the Pamir plateaus and combine this with a global dataset where seasonal Li isotope data is available to infer how hydrology may influence what we observe in the geologic record. Overall I found the

mechanism proposed reasonable, increased fluid residence time on continents associated with cooling leading to increased $d7Li$ (and vice versa for warming). That said I have numerous comments to address below before this is suitable for publication. All of this is handleable by the authors and I anticipate this work being a valuable contribution/idea if published here or elsewhere.

That said, I would like to mention that the description and analysis of the new data, shown on Ext. Data Fig. 2b could be greatly improved. If I had collected this (incredible) dataset I would feel that this paper, as written and presented, buries this new data, which in and of itself could have made a very nice paper. Given that the authors have chosen to present his work in this format I would suggest they rectify this by placing the new data in the main text if at all possible.

Major Comments

58-64: Okay yes this is how it has been cast by some of this paper's authors, but somewhere here you should re-iterate that this is not necessarily new, just a new isotope system. For example Sr isotopes, see Kump and Arthur (1997). My suggestion is that you may want to modify line 62 to be more precise and include the other isotope systems (Sr and Os), and say that these systems in sum indicate a change in "feedback strength" or "land surface reactivity", whose potential primary driver is mid-late Cenozoic tectonic uplift of major mountain ranges and the development of large flood plains in the draining river systems (and then it would be important to also cite relevant paleoaltimetry literature from Tibet, Andes, Rocky Mountains).

74: Please rephrase, especially "frustrates" with a more precise term. The fact that theory and global observations from modern rivers do not match is, in my opinion, just a reflection of the heterogeneity in landscapes available for weathering. Currently this sentence (72-75) is very opaque as to what the authors mean.

171-174: I'm fine with everything above and what has been said here in (re-)interpreting some of the isotope data, BUT I think it should be said that some of these geologic events illustrate that the modern/Neogene Li cycle, i.e. the ocean mass balance, is probably fundamentally different and had much shorter residence times in the past. This was implied by the OAE2 modeling Pogge von Strandmann, but it also just makes sense looking at the timescales shown on Figure 3. The authors also imply this, on their comment associated with OAE2 (lines 183-186), but I'm not sure why they and others seem to want to hide this.

189-194: See above, but you must be more precise here in your description of previous work. A key aspects of ref. 16 was that the $10Be/9Be$ record gives the near constant fluxes, the Li isotopes thus must infer a change in feedback strength (land surface reactivity in ref. 16). I agree, this could be due to a change in hydrology as proposed here. As the next comments suggests I think the authors are trying to provide a key distinction between their work and that of ref. 16 when such a distinguish need not be made.

196-210: This is where we differ. If there was overall drying since ~50 Ma, this could have also changed the CO_2 -weathering feedback strength relationship, and also cause $d7Li$ to increase. So I'm not sure this is an alternative scenario but I believe cohesive with previous work. If the average residence time of water on continents lengthened this would increase the feedback strength (Maher and Chamberlain, 2014 and ref 49), thus drawing down CO_2 and increasing $d7Li$. What the authors are proposing here is that hydrology not tectonics drives the $d7Li$, an interpretation that could be consistent with ref. 16.

404. As mentioned above, as also shown by ref 4. it would be more useful to show the feedback strength change implied by Caves rather than the flux. Flux is nearly constant (I can almost draw a horizontal straight line through the entire Cenozoic), what is important, as originally pointed out by Kump and Arthur (1997) and then reestablished with Li (and Be) isotopes by ref. 16 is the feedback strength change. Here and in extended figure 8 (line 610), the authors clearly are conflating weathering flux with weathering feedback strength. $10Be/9Be$ and the modeling results of flux from

ref. 4 are flux, that is different than the silicate weathering feedback strength.

Figure 1. I found this figure and analysis a bit frustrating for a few reasons. First, I assume these data do not represent the distributions but rather the highest and lowest values. This feels like cherry picking in my opinion. A stronger argument for the wet versus dry would be to re-analyze the distributions of the datasets. Split the data into wet versus dry season (need to justify this) and then look at the distributions showing they are statistically different, using a K-S test or using CDF plots. Second, The dataset seems to also be overly curated. The second author of this dataset has done extensive work on other datasets that are available and presumably show similar trends. I realize those are not as seasonality resolved, but they should somehow be represented here. Finally, in order to demonstrate that the effect on Li isotopes due to hydrology it would be helpful to see how Li concentration (or Li/Na) varies as a function of hydrology in this framework. Because it shows the 'croissant' shape (Figure 2a) it is important to do such an analysis.

Minor Line-by-Line Comments

24-25: Given that you later pitch your interpretation again ref. 16, it would be better to provide the most recent interpretation as being changes in the silicate weathering feedback strength with CO₂ via changes in land surface reactivity (which as I state above is actually in line with the author's interpretation).

27: put # of new and published measurements in parenthetical. See also below comment on 78.

51-52: Please fix grammar of this sentence, preferentially

78: The fact that all new data is from Tibet and Pamir should be added to abstract.

90-91: What are the East Mountains? I'm not sure the authors have their geography correct. The Columbia River drains the west side of the Rocky Mountains, including the Idaho Batholith, the snake river plain and cuts through the Cascade Range.

111-114: How about theoretically? It has been shown that Li isotopes have some temperature dependence with calcite precipitation. Presumably like O and H isotopes in clays the incorporation of Li should theoretically have a temperature dependence, but perhaps it is very small or overcome by kinetic effects?

159: Evidence

159-160: Plant leaf wax dD?

377. Figure 2 - are these all averages or individual measurements?

391: Are the geologic examples here corrected for the fractionation between water and calcite? From 459-467 this does not appear to be the case.

392: Error bars on points, see comment on Figure 1 above

394: c) it should be noted that these are non-marine, because of the residence time of Li on the ocean this initially confused me as being implausible (having forgotten about these data)

550. This figure is very messy, I would break up the panels. Additionally, looking at this data, only the BH River and Yellow River really have the datasets that I would be comfortable picking a max and min value from (see my comments on Figure 1 above)

553: "discharges" should be "river discharge"

557: "keeps" → "stays"

559-562: Most things on Wikipedia themselves have primary references, using wikipedia as the data source seems too secondary to me.

Review by Dr. Daniel E. Ibarra, UC Berkeley/Brown University

Reviewer #3 (Remarks to the Author):

Overarching comments:

The paper presents Li isotopic data from rivers collected over seasonal shifts in discharge and based on the differences in isotopic composition, the authors argue that residence time is responsible for the observed Cenozoic shifts in Li isotopes. I liked the first part of the paper and also found the middle synthesis section, comparing to various marine and speleothem records, interesting. That said, I think there are several important limitations of the current analysis that need to be addressed.

First, is the issue of Li isotopes vs Li fluxes. The authors should include Li fluxes and the flux-weighted Li isotopic composition in their analysis. It is the flux-weighted Li isotopic composition that matters for interpreting marine records and hence one cannot simply compare across basins. As noted in point 2, this also bears strongly on the interpretation of the sensitivity of Li to residence time. I suspect the flux-weighted Li isotopic composition and Li fluxes for most rivers will be highly dominated by the high discharge events, meaning what is actually transmitted to oceans is the “wet season” signal. For example, the SL river: discharge varies by ca. 2.5 orders of magnitude between dry and wet and presumably Li is diluted and concentrations are lower at high discharge. What fraction of yearly Li flux arises from wet vs dry? I suspect the periods of high discharge dominate the Li mass flux and isotopic composition so I think this needs to be evaluated. It would be interesting to show the $d7Li_{dry_to_wet}$ vs the flux-weighted $d7Li$ since the latter is ultimately what matters for interpreting the oceanic Li record. Consequently, it is very hard to evaluate the paper without seeing the results of this analysis.

Second are issues of residence time. The Sr isotope data presented seems to contradict the argument the authors make, but seemingly the Sr isotope data agree more with current models of runoff and solute generation in watersheds, where the seasonal variations in discharge are associated with shifting distributions of flow paths, from baseflow in dry periods, to increasing interflow/surface runoff at high discharge. So as above, the key question is does baseflow (groundwater, likely with longer residence time and in contact with fresher regolith) or high discharge (mixing of multiple shallow water sources with baseflow) set the Li isotopic composition? Regardless, the apparent contradiction between the Sr data, which show changing flow paths, and the author's interpretation of Li as a controlled by residence time, need to be reconciled.

Relatedly, residence time and reactivity are somewhat interchangeable for weathering and isotopes. Longer residence times can in theory produce the same result as more reactive regolith with shorter residence times. In both cases, there is more unweathered parent material to convert to clay minerals, so you expect a higher $d7Li$. This is likely why baseflow/groundwater during low discharge has high $d7Li$ – water circulates deeper and sees fresh regolith. However, over the million timescales relevant for the oceans, it could be reactivity or residence time and it is not apparent to me how one would distinguish. I also did not see a clear basis for neglecting reactivity outside of the PETM/OAE records, particularly when one looks at the Cenozoic.

Residence time also does not scale in a straightforward fashion with climate as the volume of water in storage also declines. The authors appear to have the data to calculate residence time with some assumptions, as they have yearly discharge and basin size. Given that they don't see correlation with basin size, it is not clear what this would like relative to figure 2b, but likely the assumptions of depth and porosity are potentially moderate to small. If the authors are actually thinking of travel time instead of residence time, then they do need to acknowledge the issue of flow paths and water sourcing in analysis 1 – the issue of flux weighting and water sourcing within the hydrograph.

Third, I think the discussion of hydrologic control on Li isotopes is useful but if find the heavy reliance on the geologic records, in absence of the flux-weighted analysis and a mechanistic discussion, a bit challenging to believe. Residence times, driven by an attendant decrease in discharge, will decrease weathering fluxes so this explanation for Li isotopic shifts also needs to maintain balance in the

carbon cycle and it is not clear that it can. In figure 4, for example, the $\delta^7\text{Li}$ increases in the late Miocene, while weathering fluxes are somewhat invariant. So the key question to me is: How substantial of a shift in hydrology would it take to drive the $\delta^7\text{Li}$ transitions and why is that not reflected in the weathering flux? I think this question needs to be addressed.

In summary, I appreciated the first discussion and the seasonality in the Li isotope data is interesting in useful, especially if reconciled with the Sr data. I struggled to be convinced by the second argument, about global implications: I think the authors do have an argument about the importance of hydrology for Li, but residence time is likely just one of many factors, and any effects on hydrology have to be considered in light of weathering fluxes and carbon cycle balance. Compared to invoking increased reactivity, that allows for greater sensitivity, the change in discharge invoked here seems harder to reconcile.

A few detailed comments:

-There are a lot of very minor grammatical or typographical corrections that should be made but I assume they would be addressed at a later stage and did not mark them below.

Abstract: Line 22 – awkward phrasing - clarify what “composition” is being referred to.

Line 84: It is not just residence time, but the distribution of residence times and location of flow paths, e.g., flushing of shallow soils during spring snowmelt are well-known from variations in concentration-discharge relationships.

Figure 1: I was a little confused by the presentation of sites – can the authors clarify this or maybe consider alternative presentation?

Line 93: 20% of freshwater but what fraction of Li? Generally, rivers in the tropics are ~80% of global discharge, and presumably global Li flux?

Line 102: The fact that the Sr shifts indicates different water sourcing within a yearly hydrography – evidence not simply a straightforward residence time the authors imply?

Line 106: I haven't read all these papers on residence time for Li, but I would guess many of them look at 1-D systems not 3-D watersheds? The authors at line 106 need to clarify mixing – a hydrograph for a river is itself a manifestation of mixing – different fractions of groundwater, interflow and surface runoff. Usually, low discharge is predominated by groundwater so leading to line 108, given that most rocks are the same composition, I don't understand this argument at all. You would need to look at Li mass balance.

Line 116: the authors cite a study for long and short residence times based on dry and wet seasons, which is intuitive, but this particular study looks at 22 Swiss catchments – for larger scale analysis/models, like continental US, I believe the picture is more complicated? Anyway, I agree with the general interpretation, but it would be nice to backstop it with basic catchment hydrology – generally old groundwater as baseflow vs. interflow vs. surface runoff, where the latter two supplement or dilute the baseflow at high discharge.

Figure 2: is this all the data or are some averaged? I some of these are averaged or one sample it seems a little confusing to plot them if the point is the seasonal data. I was a little confused about what was plotted. In addition, is it really fair to draw a line from 3 averages??? Finally, an important point for the latter arguments in the paper: is it the flux of Li that matters and if one plots $\delta^7\text{Li}$ vs Li flux, what does that look like? Active mountains and tropical lowlands have very different Li concentrations – might be a nice additional panel.

I like the paper from intro to about line 143, although I think the Sr issue and hydrology of basins could be better represented, though I appreciate the desire to keep the argument simple.

Figure 3. I am not really sure the Hirnantian glaciation is worth including given how different weathering was at that time and how uncertain the records are, and generally it doesn't add much. The Cenozoic and Cretaceous records make sense. For the speleothem, visually it might be easier to highlight the glacial in red to be consistent with other records, it is confusing they are blue but Li values go up, then delete the Hirnantian record.

Line 145: I like this discussion – it is an important piece to address the question of how seasonal variations observed in the first part may or may not translate into LI associated with mean climate states. I think this point should be made more strongly at the beginning, especially because the riverine data could strongly reflect one season once you flux weight it. That said, I don't much about the fidelity or current interpretations of the PETM and OAE marine records and hopefully another review can comment.

Line 182: Again, I worry about this conclusion... residence time doesn't scale perfectly with climate-driven hydrologic changes and during these times there were other variables, such as topography, vegetation, etc. that change residence time. For example, residence time (assuming the authors mean residence time and not travel time) is defined as volume of water in storage divided by the discharge – during drier periods water tables are also potentially lower so less water in storage means that residence time could potentially be unchanged. Finally, if you change residence time you likely change the weathering flux, concentrations go up but discharge goes down, and thus given that weathering fluxes scale strongly with discharge it is hard to see how this can balance out.

Line 194-202: Can this be broken down a little bit? Hard to follow.

Line 204: I think the authors present evidence for a hydroclimate signal in the Li isotopes – residence time is just one of the potential explanations (or part of a more comprehensive one) so personally I think this is perhaps stepping beyond the evidence presented?

Below, we have copied all comments from the reviewers (*black, italics*) followed by our responses in blue text. We have revised the text (red) in the manuscript accordingly, and provided the revised manuscript with line numbers (RML). We thank the reviewers for the opportunity to greatly improve our manuscript.

Reviewer #1(Dr. Sambuddha):

It was a pleasure reading the manuscript by Zhang et al. In my opinion it's one of the most lucid manuscripts I have read in a while. The work presented here utilizes published data in association with new time-series data to provide new insights into what controls the lithium isotopic composition of river water. Lithium concentration and isotopic composition of river water has been widely applied to quantify the rate of terrestrial silicate weathering and thus, the rate of CO₂ drawdown by weathering. This work proposes a new explanation for the long-term (Cenozoic) changes in seawater lithium isotopic composition to better constrain the role of silicate weathering in drawing down atmospheric CO₂ concentration over this period to usher in modern climate. I would recommend this manuscript for publication with a few key changes / additions.

Re: First, we appreciate Reviewer #1 for thinking that our study provides a new explanation for long-term seawater $\delta^7\text{Li}$ records that would better constrain past continental weathering and the carbon cycle. Second, we also appreciate his thoughtful and constructive comments that follow. In our view, all these comments have helped to improve the manuscript considerably by adding clarity to the interpretations and discussion. In particular, the suggestion of variable seawater Li residence time enables us to connect modern dataset to long-term marine Li records.

The major comments are the following:

Firstly, when comparing historical $\delta^7\text{Li}$ record of seawater it is risky to include data from studies that utilized bulk carbonates. We now know that different carbonate archives have a different absolute $\delta^7\text{Li}$ value when grown in the same seawater matrix (probably vital effect). Seawater $\delta^7\text{Li}$ excursions beyond the foraminifera archive is reconstructed from bulk carbonates and focuses on events of abrupt climate change. My strong recommendation is to exclude / minimize the discussion on parts of the paleo seawater $\delta^7\text{Li}$ record based on bulk carbonates.

Re: We agree with Reviewer #1 that different carbonate archives could have a different absolute $\delta^7\text{Li}$ value, probably for the vital effects they note. Because of this reason, the authors of the original publications discussed Li isotopic fractionation factors for seawater carbonate precipitation in the OAE1a, OAE2 and Hirnantian glaciation events (Lechler et al., 2015; Pogge von Strandmann et al., 2013, 2017a), and based on reported experimental data, they proposed an $\sim 3-5\%$ lower $\delta^7\text{Li}$ in bulk carbonate than those in seawater. They then suggested that this range is small compared to the $\delta^7\text{Li}$ variation during these climate events. While this is important, it is not in the scope of our study to address these complexities, and they do not have a major impact on the interpretations put forth in our study.

In more detail, (1) although there are uncertainties on the fractionation factor, we do not seek to compare the absolute seawater $\delta^7\text{Li}$ values between events. What we are really interested in is their *relative changes* through time in each individual event, not the absolute values. The large change of bulk carbonate $\delta^7\text{Li}$ trend in an individual event is sufficient to discuss its climatic effects, e.g., as the climate was wetter in the OAE 2, the $\delta^7\text{Li}$ was lower, and in the drier climate in the Hirnantian glaciation, there was higher $\delta^7\text{Li}$; (2) We noted that the $\delta^7\text{Li}$ values in bulk carbonate were indeed very similar with the independent brachiopod samples for the Hirnantian event (Fig. 4g); (3) A new carbonate-based Li isotope record, spanning the past 3.0 billion years, indicates that an increase in carbonate Li isotopes over time was driven by long-term changes in the Li isotopic conditions of sea water, rather than in the sedimentary alteration of older samples. The massive dataset (>600 carbonate samples from 101 stratigraphic units) supports a reliable seawater Li archive reconstructed by bulk carbonate—indicating that bulk carbonates can be used to faithfully reconstruct seawater $\delta^7\text{Li}$ in the deeper past (Kalderon-Asael et al., 2021, Nature). Based on this, they reproduce previously estimated seawater $\delta^7\text{Li}$ values during the Cenozoic as well as the Mesozoic and the Palaeozoic, assuming an offset of $4 \pm 5\%$ from sea water. We have added this result in the Method section.

For these reasons, it is reasonable to discuss the paleo seawater $\delta^7\text{Li}$ record. We thank Reviewer#1 for his professional suggestions to make us to be more cautious about these data. The reason we hope to keep this aspect of our paper is that these 3 paleo-records provide scarce but valuable evidence to confirm that short-lived extreme climate events can also cause significant changes in seawater $\delta^7\text{Li}$, and particularly, their variabilities are comparable or even greater than the long-term Cenozoic changes (9‰). This could have meaningful implication for a large number of scientists working on palaeo-event studies. We hope that this explanation will satisfy Reviewer #1.

Secondly, the impact of a variable seawater Li residence time on the interpretation drawn by the authors should be incorporated in the manuscript. We are dealing with a classical scenario of changes in river flux and composition vs. seawater $\delta^7\text{Li}$ composition. The rate of change of $\delta^7\text{Li}$ of seawater is expected to be a direct function of Li residence time. When dealing with a long-term record sticking to modern day residence time estimates should be avoided. I would recommend that the authors include a variable Li residence time in their discussion.

Re: Great suggestions. Reviewer #2 also suggest that “some of these geologic events illustrate that the modern/Neogene Li cycle, i.e. the ocean mass balance, is probably fundamentally different and had much shorter residence times in the past. This was implied by the OAE2 modeling Pogge von Strandmann, but it also just makes sense looking at the timescales shown on Figure 4”. This is a nice comment to improve our data interpretation. Following Reviewer#1’s suggestion, we have clarified residence times in the river systems and in the ocean to avoid confusion, by adding a box model to discuss variable seawater Li residence time (see Supplementary Note 2). We have also added discussions in main text. (RML: 195-202, 234-242)

Thirdly, the impact of clay cannibalization (as demonstrated on Rio Negro samples by co-author Dellinger) should be discussed in detail in the manuscript. This process has the ability to strongly impact the $\delta^7\text{Li}$ of tropical rivers during low flow (characterized by long residence time of water). Dellinger established that the impact of this process will influence the $[\text{Li}]$ and $\delta^7\text{Li}$ of the dissolved phase in a way identical to that of congruent weathering of rocks. I surprised that this aspect of Li fluvial chemistry was completely omitted from the manuscript.

Re: Yes, this is a reasonable concern about clay cannibalization. The reasons we didn't discuss clay cannibalization include: 1) It is still unclear how important the congruent or incongruent dissolution of tropical clays will affect riverine $\delta^7\text{Li}$. The published data showed that tropical low-relief rivers could produce both low (organic-rich blackwater) and high (lateritic settings) $\delta^7\text{Li}$ values, e.g., lowland rivers in the Congo and Amazon basins (typical of a supply-limited weathering regime) (Dellinger et al., 2015; Henchiri et al., 2016). This means that in addition to clay congruent dissolution in blackwater environments, there are also tropical lateritic regions that are still pedogenically active and secondary weathering products are forming there; 2) Importantly, even in the Negro River, we observed a consistent hydrological dependence of riverine $\delta^7\text{Li}$ (Fig. R1). This suggests that in extremely weathered environments, riverine $\delta^7\text{Li}$ still has a hydrological dependence; 3) Two data for seasonal $\delta^7\text{Li}$ in the Fig.R1 can be freely downloaded online from a Ph. D thesis, but not yet published. We have contacted the authors, and they plan to publish a paper later. Based on academic ethics, we did not include these data in our text.

Fig. R1 | Seasonal variations of $\delta^7\text{Li}$ in the extremely weathered Negro River.
(Date in March and May from Dellinger et al., 2015; Data in June and November from Fries, 2018, Ph. D thesis)

There is no mention of possible ground water contribution in controlling $\delta^7\text{Li}$ of rivers during low flow – this must be touched upon. There is a limited data on $\delta^7\text{Li}$ of ground water; however, recent work from Josh West's group have put realistic constrains on the $\delta^7\text{Li}$ of ground water in Bengal basin. I would recommend the authors utilize that study to discuss possible influence of ground water on riverine Li chemistry.

Re: Added discussion of potential influence of ground water on riverine Li chemistry and also the reference in our main text as suggested, thanks. (RML: 137-141)

Reviewer #2 (Dr. Daniel E. Ibarra, UC Berkeley/Brown University):

Summary

Zhang and coauthors provide new high-temporal resolution river Li isotope data from Tibet and the Pamir plateaus and combine this with a global dataset where seasonal Li isotope data is available to infer how hydrology may influence what we observe in the geologic record. Overall I found the mechanism proposed reasonable, increased fluid residence time on continents associated with cooling leading to increased $d7Li$ (and vice versa for warming). That said I have numerous comments to address below before this is suitable for publication. All of this is handleable by the authors and I anticipate this work being a valuable contribution/idea if published here or elsewhere.

Re: We appreciate Reviewer #2 for supporting our proposed mechanism, and thinking our work will be a valuable contribution. In particular, the constructive suggestion of placing our new time-series data in the main text helps greatly strengthen the understanding of residence times in modern rivers, and the suggestions on feedback strength further enhance our in-depth understanding of global weathering and the carbon balance.

That said, I would like to mention that the description and analysis of the new data, shown on Ext. Data Fig. 2b could be greatly improved. If I had collected this (incredible) dataset I would feel that this paper, as written and presented, buries this new data, which in and of itself could have made a very nice paper. Given that the authors have chosen to present his work in this format I would suggest they rectify this by placing the new data in the main text if at all possible.

Re: Thanks for the high praise of our new dataset. Following the suggestion, we have placed the new data in the main text. (RML: 87-106)

Major Comments

58-64: Okay yes this is how it has been cast by some of this paper's authors, but somewhere here you should re-iterate that this is not necessarily new, just a new isotope system. For example Sr isotopes, see Kump and Arthur (1997). My suggestion is that you may want to modify line 62 to be more precise and include the other isotope systems (Sr and Os), and say that these systems in sum indicate a change in "feedback strength" or "land surface reactivity", whose potential primary driver is mid-late Cenozoic tectonic uplift of major mountain ranges and the development of large flood plains in the draining river systems (and then it would be important to also cite relevant paleoaltimetry literature from Tibet, Andes, Rocky Mountains).

Re: Revised as suggested, thanks. (RML: 62-64)

74: Please rephrase, especially "frustrates" with a more precise term. The fact that theory and global observations from modern rivers do not match is, in my opinion, just a reflection of the heterogeneity in landscapes available for weathering. Currently this sentence (72-75) is very opaque as to what the authors mean.

Re: We have rephrased this sentence, thanks. (RML: 72-75)

171-174: *I'm fine with everything above and what has been said here in (re-)interpreting some of the isotope data, BUT I think it should be said that some of these geologic events illustrate that the modern/Neogene Li cycle, i.e. the ocean mass balance, is probably fundamentally different and had much shorter residence times in the past. This was implied by the OAE2 modeling Pogge von Strandmann, but it also just makes sense looking at the timescales shown on Figure 3. The authors also imply this, on their comment associated with OAE2 (lines 183-186), but I'm not sure why they and others seem to want to hide this.*

Re: Accepted. This is a nice suggestion to deepen our conclusions and is also helpful to answer the concern from Reviewers #1 and #3. We have added to our text. (RML: 197-202)

Following this suggestion, we also find that there are papers suggesting a similar view: 1) Explaining the light Li isotopes require the riverine flux to increase 2 to 4 times, and a lower oceanic Li residence time during the OAE 2 relative to the present day (Jenkyns et al., 2017; Pogge von Strandmann et al., 2013); 2) For the relatively short time of recovery, seawater Li residence time during the OAE 1 is estimated to be about 1/3 of today's (Lechler et al., 2015).

189-194: *See above, but you must be more precise here in your description of previous work. A key aspects of ref. 16 was that the $^{10}\text{Be}/^{9}\text{Be}$ record gives the near constant fluxes, the Li isotopes thus must infer a change in feedback strength (land surface reactivity in ref. 16). I agree, this could be due to a change in hydrology as proposed here. As the next comments suggests I think the authors are trying to provide a key distinction between their work and that of ref. 16 when such a distinguish need not be made.*

Re: Thanks, we have added feedback strength to describe the previous work as suggested (RML: 224-228). Based on our data of the influence of hydrology on $\delta^7\text{Li}$ in the present-day, we suggest that the Cenozoic seawater $\delta^7\text{Li}$ rise is the result of an increase of the fluid residence time on the continents due to an overall drying global climate. This is consistent with the declining continental precipitation from the Pacific and Atlantic sides of Eurasia during the last 50 Ma (Fig. 5 d-e). Therefore, we think that the reduction of the hydrological cycle instead/in addition to the increases of erosion related to mountain building, leads to a strengthening of the weathering feedback, finally resulting in a roughly stable weathering flux despite lower $p\text{CO}_2$. The Li isotope results across a large range of temporal and spatial scales confirm the picture of past hydrological changes and further improve the understanding the feedback strength on continental weathering. On this point, our conclusion is now in line with the ref.16 (Caves Rugenstein et al., 2019).

196-210: *This is where we differ. If there was overall drying since ~50 Ma, this could have also changed the CO_2 -weathering feedback strength relationship, and also cause $\delta^7\text{Li}$ to increase. So I'm not sure this is an alternative scenario but I believe cohesive with previous work. If the average residence time of water on continents lengthened*

this would increase the feedback strength (Maher and Chamberlain, 2014 and ref 49), thus drawing down CO₂ and increasing $\delta^7\text{Li}$. What the authors are proposing here is that hydrology not tectonics drives the $\delta^7\text{Li}$, an interpretation that could be consistent with ref. 16.

Re: We thank the reviewer for the clarification and the suggestion. We agree that our interpretation is consistent with ref.16, i.e., that an increase in the average residence time of water on continents would increase the feedback strength and the $\delta^7\text{Li}$ values, without a significant change of the weathering flux, in line with the Be isotope record interpretation from refs.4 and 16 (Caves Rugenstein et al., 2016, 2019).

However, we also note that in our compiled dataset (Supplementary Fig.11c), lowland rivers from dry climates with high $\delta^7\text{Li}$ (long water-rock contact time) show lower weathering rates than lowland rivers with wet climate and low $\delta^7\text{Li}$ (short water-rock contact time). Therefore, we think there are two possibilities: i) that a drying climate with decreasing continental runoff would result in a decreased weathering and CO₂ drawing down, rather than increase in weathering feedback. This would agree with other new publications suggesting, based on different proxies, a decrease in weathering fluxes during the Cenozoic (Clift and Jonell, 2021; Si and Rosenthal, 2019); and ii) that the increase in feedback strength caused by mid-late Cenozoic uplift (such as Caves Rugenstein et al., 2016, 2019) could lead to an increase in weathering, and this could compensate for the decreasing weathering flux due to cooling and drying (Kump and Arthur, 1997). These two combined effects would compensate each other and lead to a roughly stable late Cenozoic weathering flux. (RML: 252-281)

404. As mentioned above, as also shown by ref 4. it would be more useful to show the feedback strength change implied by Caves rather than the flux. Flux is nearly constant (I can almost draw a horizontal straight line through the entire Cenozoic), what is important, as originally pointed out by Kump and Arthur (1997) and then reestablished with Li (and Be) isotopes by ref. 16 is the feedback strength change. Here and in extended figure 8 (line 610), the authors clearly are conflating weathering flux with weathering feedback strength. $^{10}\text{Be}/^{9}\text{Be}$ and the modeling results of flux from ref. 4 are flux, that is different than the silicate weathering feedback strength.

Re: We have added the feedback strength in Fig.5 as suggested. The decreasing hydrology could decrease the weathering flux, while at the same time increase reactivity by late Cenozoic uplift. These two effects finally lead to a stable weathering flux. By adding these discussions, we think that our revision now introduced reasonable evidence to highlight the hydrology role on controlling Li isotopes and continental weathering, and our conclusion is consistent with a change of the feedback strength. Please see also the replies above.

Figure 1. I found this figure and analysis a bit frustrating for a few reason. First, I assume these data do not represent the distributions but rather the highest and lowest values. This feels like cherry picking in my opinion. A stronger argument for the wet versus dry would be to re-analyze the distributions of the datasets. Split the data into wet versus dry season (need to justify this) and then look at the distributions showing

they are statistically different, using a K-S test or using CDF plots.

Second, The dataset seems to also be overly curated. The second author of this dataset has done extensive work on other datasets that are available and presumably show similar trends. I realize those are not as seasonality resolved, but they should some how be represented here. Finally, in order to demonstrate that the effect on Li isotopes due to hydrology it would be helpful to see how Li concentration (or Li/Na) varies as a function of hydrology in this framework. Because it shows the 'croissant' shape (Figure 2a) it is important to do such an analysis.

Re: Thanks for the suggestions. First, some rivers only have a few time-series/seasonal data points, and so have insufficient data to do distribution analysis. In addition, the wet versus dry seasons are defined either by the original authors or by the water discharge, rather than determined by the $\delta^7\text{Li}$ values., e.g., we checked the seasonal sampling date, and then found that higher $\delta^7\text{Li}$ were corresponded to dry seasons, and lower values to wet seasons. Even then, following the Reviewer's suggestions, we have also added a figure to re-analyze the datasets and they do show differences between dry and wet seasons (Supplementary Fig. 6).

Second, we have compiled all seasonal data carefully. We didn't add the datasets of our second author in our first submission, because the seasonal sampling times were too close to differentiate the dry or wet seasons (Supplementary Fig. 7), and so cannot compare their seasonal variations. Following the suggestions, we reconsidered this dataset, and think it may also be helpful to enhance our conclusion. The dataset shows similar runoff having similar $\delta^7\text{Li}$ in the Amazon River systems, which further supports a hydrological dependence of river $\delta^7\text{Li}$.

Finally, we have added $\delta^7\text{Li}$ versus Li/Na ratios for the spatial dataset in Fig. 3a as suggested. We also added runoff versus Li concentration for all time-series data (Supplementary Fig.13a), which showing a dilute effect.

Minor Line-by-Line Comments

24-25: Given that you later pitch your interpretation again ref. 16, it would be better to provide the most recent interpretation as being changes in the silicate weathering feedback strength with CO₂ via changes in land surface reactivity (which as I state above is actually in line with the author's interpretation).

Re: Added as suggested. (RML: 23-24)

27: put # of new and published measurements in parenthetical. See also below comment on 78.

Re: Revised. (RML: 27)

51-52: Please fix grammar of this sentence, preferentially

Re: Revised. (RML: 53)

78: The fact that all new data is from Tibet and Pamir should be added to abstract.

Re: Added as suggested. (RML: 27)

90-91: *What are the East Mountains? I'm not sure the authors have their geography correct. The Columbia River drains the west side of the Rocky Mountains, including the Idaho Batholith, the snake river plain and cuts through the Cascade Range.*

Re: Thanks for the reminder. We have changed it to “the east of the Cascades” (RML: 113)

111-114: *How about theoretically? It has been shown that Li isotopes have some temperature dependence with calcite precipitation. Presumably like O and H isotopes in clays the incorporation of Li should theoretically have a temperature dependence, but perhaps it is very small or overcome by kinetic effects?*

Re: Yes, it is very small. A simple temperature dependence of the clay fractionation factor during weathering would only cause <1.7‰ variation (Li and West, 2014; Pogge von Strandmann et al., 2017b).

159: *Evidence*

Re: Revised. (RML: 185)

159-160: *Plant leaf wax dD?*

Re: Revised. (RML: 185)

377. *Figure 2 - are these all averages or individual measurements?*

Re: For these spatial rivers, existing $\delta^7\text{Li}$ data are overall scarce. For spot sample with multiple measurements by different studies, we calculated an average value, and others are individual measurements. We have clarified it in the supplementary Data 2.

391: *Are the geologic examples here corrected for the fractionation between water and calcite? From 459-467 this does not appear to be the case.*

Re: Not corrected. Generally, the original authors used a fractionation factor of 3-5‰ between calcite and seawater, and as suggested by Reviewer#1, there are uncertainties. Here, we do not intend to compare absolute seawater $\delta^7\text{Li}$ between events. The large change in individual events is sufficient for us to discuss the trend, so we do not correct the data.

392: *Error bars on points, see comment son Figure 1 above*

Re: Thanks, we have added “Errors for $\delta^7\text{Li}$ are similar to the symbol size” to the legend of the Figures.

394: *c) it should be noted that these are non-marine, because of the residence time of Li on the ocean this initially confused me as being implausible (having forgotten about these data)*

Re: Yes, we have noted it in both Fig. 4 and its legend, thanks.

550. *This figure is very messy, I would break up the panels. Additionally, looking at this*

data, only the BH River and Yellow River really have the datasets that I would be comfortable picking a max and min value from (see my comments on Figure 1 above)

Re: Accepted. We have moved the BH and SL to main text (Fig.1), and also separated the figure as Supplementary Figs.5 & 10 as suggested. Please see reply to the comments on Fig.1.

553: “discharges” should be “river discharge”

Re: Revised as suggested.

557: “keeps” —> “stays”

Re: Revised. Thanks.

559-562: *Most things on Wikipedia themselves have primary references, using wikipedia as the data source seems too secondary to me.*

Re: Revised as suggested. We appreciate the careful review.

Reviewer #3:

Overarching comments:

The paper presents Li isotopic data from rivers collected over seasonal shifts in discharge and based on the differences in isotopic composition, the authors argue that residence time is responsible for the observed Cenozoic shifts in Li isotopes. I liked the first part of the paper and also found the middle synthesis section, comparing to various marine and speleothem records, interesting. That said, I think there are several important limitations of the current analysis that need to be addressed.

Re: We appreciate Reviewer#3 for his interest in our manuscript. First, we have considered carefully for the constructive comments below, and particularly, more data analysis/discussion have been added for the Reviewer#3's most concern question--Li fluxes. Second, we have added newly measured data for time-series Sr isotopes to clarify the controlling mechanisms of Li-Sr isotopes. Third, we have also added detailed discussions regarding weathering flux and the carbon mass balance. Finally, after adding the discussions, our manuscript has been greatly improved, and the conclusions are more robust now. Here, we appreciate to have an opportunity to clarify that this study cannot tackle all the detailed aspects of hydrological processes and their effects on Li isotopes, but our study aims more at highlighting the potentially underestimated role of hydrology on $\delta^7\text{Li}$ and opens the door for future more detailed studies.

First, is the issue of Li isotopes vs Li fluxes. The authors should include Li fluxes and the flux-weighted Li isotopic composition in their analysis. It is the flux-weighted Li isotopic composition that matters for interpreting marine records and hence one cannot simply compare across basins. As noted in point 2, this also bears strongly on the interpretation of the sensitivity of Li to residence time. I suspect the flux-weighted Li isotopic composition and LI fluxes for most rivers will be highly dominated by the high discharge events, meaning what is actually transmitted to oceans is the “wet season” signal. For example, the SL river: discharge varies by ca. 2.5 orders of magnitude

between dry and wet and presumably Li is diluted and concentrations are lower at high discharge. What fraction of yearly Li flux arises from wet vs dry? I suspect the periods of high discharge dominate the Li mass flux and isotopic composition so I think this needs to be evaluated. It would be interesting to show the $d7Li_{dry_to_wet}$ vs the flux-weighted $d7Li$ since the latter is ultimately what matters for interpreting the oceanic Li record. Consequently, it is very hard to evaluate the paper without seeing the results of this analysis.

Re: We appreciate and totally agree with the thoughtful comments for the Li flux. According to the suggestions and after careful consideration, a more suitable Li yield analysis was used to interpret marine records, because this enables us to compare the Li fluxes of all compiled river data under a same unit area. For spatial rivers across latitudes, the new modified figure shows that high runoff results in a high Li yield with low δ^7Li (Fig. 3b-c). For seasonal data, we used high-resolution daily Li yield to compare both the changes in individual basin and among basins (Supplementary Fig.13). The results show that although high runoff diluted Li concentrations, but it remains resulting in a high Li yield with low δ^7Li . Here, we cannot analyze the flux-weighted δ^7Li between dry vs. wet seasons for global rivers, because for many rivers, the data of seasonal δ^7Li and/or river discharge are still unavailable. Despite this, both the spatial and seasonal Li yields provide valuable information and are consistent with Reviewer#3's expectation.

In terms of interpreting the oceanic Li records, one important aspect is that if the global hydrological cycle has slowdown during the Cenozoic, it means the overall global runoff has decreased and water residence time has increased. This would result in a higher contribution of Li from dry seasons (with higher δ^7Li) and/or a shift in the annual average (higher δ^7Li of both wet and dry seasons). In both cases, the Li isotopic compositions of global rivers would increase. It is clear in Fig. 3 and Supplementary Fig. 13, that wet climate, characterized by high runoff, would result in high river Li flux and low δ^7Li to the oceans, whereas dry climate would produce low Li flux and high δ^7Li to the oceans. A long-term hydrological shift from wet to dry climate would further lengthen the seawater Li residence times and thereby increase the δ^7Li (Supplementary Note 2 and please see also the reply to Reviewer #1). Thus, by considering Li flux and seawater residence time, as suggested by reviewers#1 and #3, we provide more evidence to support our new explanation for changes in seawater δ^7Li over the Cenozoic.

Second are issues of residence time. The Sr isotope data presented seems to contradict the argument the authors make, but seemingly the Sr isotope data agree more with current models of runoff and solute generation in watersheds, where the seasonal variations in discharge are associated with shifting distributions of flow paths, from baseflow in dry periods, to increasing interflow/surface runoff at high discharge. So as above, the key question is does baseflow (groundwater, likely with longer residence time and in contact with fresher regolith) or high discharge (mixing of multiple shallow water sources with baseflow) set the LI isotopic composition? Regardless, the apparent contradiction between the Sr data, which show changing flow paths, and the author's interpretation of Li as a controlled by residence time, need to be reconciled.

Re: The reason for comparing the Li-Sr isotopes is that we noted irregular seasonal characters of global river water $^{87}\text{Sr}/^{86}\text{Sr}$ (Supplementary Fig. 9). But after analyzing the seasonal $\delta^7\text{Li}$ of the BH and SL rivers (Fig. 1), and further extending to other global rivers, we observed an amazing consistent pattern in $\delta^7\text{Li}$ seasonality (Fig. 2). This reveals a great potential for Li isotopes to link to hydrological cycles.

We think that Sr and Li isotopes are controlled by two distinct mechanisms. Different rock/soil minerals have distinct Sr isotopic compositions, but similar Li isotopes for silicates. So, during the weathering processes, fluid $^{87}\text{Sr}/^{86}\text{Sr}$ ratios are controlled by the types of rocks/minerals it contacted. In contrast, fluid $\delta^7\text{Li}$ are controlled by the rates of secondary mineral formation, with minor impact from mineral types.

We agree with the Reviewer that different seasons have different flow paths and residence times. Fundamentally, changes in river $^{87}\text{Sr}/^{86}\text{Sr}$ from dry to wet seasons are determined by changes in mineral types that water contacted, due to varying flow paths, water sources, etc. However, changes in $\delta^7\text{Li}$ during seasonal shifts are still determined by the rates of secondary mineral formation, rather than mineral types, though the flow paths are changed. Therefore, fast surficial flows in rain seasons (short water-rock contact times, and low rates of secondary mineral formation) show commonly low $\delta^7\text{Li}$ values relative to dry seasons in our dataset, while $^{87}\text{Sr}/^{86}\text{Sr}$ (determined by the changing flow path and thus varying mineral types that it contacted) has either high or low values as shown for global basins (Supplementary Fig. 9).

To better explore their differences, we have measured new Sr isotope data for the two Tibetan rivers (BH and SL, Fig.1). The data show consistent hydrology dependence of $\delta^7\text{Li}$, but $^{87}\text{Sr}/^{86}\text{Sr}$ are largely different.

Together, for seasonal variations, high water discharge set Li fluxes with low $\delta^7\text{Li}$ as shown in supplementary Fig. 13; for long-time climate changes, a drying Cenozoic climate decreases runoff and Li flux but increase $\delta^7\text{Li}$ from continental rivers to the oceans, similar as dry seasons (Fig.3 and Supplementary Fig.13).

Relatedly, residence time and reactivity are somewhat interchangeable for weathering and isotopes. Longer residence times can in theory produce the same result as more reactive regolith with shorter residence times. In both cases, there is more unweathered parent material to convert to clay minerals, so you expect a higher $\delta^7\text{Li}$. This is likely why baseflow/groundwater during low discharge has high $\delta^7\text{Li}$ – water circulates deeper and sees fresh regolith. However, over the million timescales relevant for the oceans, it could be reactivity or residence time and it is not apparent to me how one would distinguish. I also did not see a clear basis for neglecting reactivity outside of the PETM/OAE records, particularly when one looks at the Cenozoic.

Re: Yes, high reactivity with shorter residence times in theory can produce a same result as longer residence time. But if we see the river $\delta^7\text{Li}$ values between the tectonically active mountains (25 rivers) and the dry mid-high latitude lowlands (27 rivers) (Fig. 3a-b), obviously lower $\delta^7\text{Li}$ are found in the mountain areas where reactivities are high. As the mountain dataset covers broad regions, e.g., the Andes, the Himalayas-Tibetan Plateau, the New Zealand Alps, and the Rocky and the Machenzie Mountains, this

implies that relative to land surface reactivity, residence time linked to hydrology is still a major control on riverine $\delta^7\text{Li}$ as we proposed.

Residence time also does not scale in a straightforward fashion with climate as the volume of water in storage also declines. The authors appear to have the data to calculate residence time with some assumptions, as they have yearly discharge and basin size. Given that they don't see correlation with basin size, it is not clear what this would like relative to figure 2b, but likely the assumptions of depth and porosity are potentially moderate to small. If the authors are actually thinking of travel time instead of residence time, then they do need to acknowledge the issue of flow paths and water sourcing in analysis 1 – the issue of flux weighting and water sourcing within the hydrograph.

Re: Thanks. By carefully considering the Reviewer's comments, we have added newly measured time-series Sr isotopes for the Tibetan rivers to differentiate the possible water sources/flow paths (Fig.1). These provide typical cases to clarify this question. We also added Li flux analysis for spatial and seasonal dataset across latitudes (Fig.3 and Supplementary Fig. 13). In addition, we added variable seawater Li residence time discussions (Supplementary Note 2 and Table 1). These analyses and discussions now further support our residence time idea.

Third, I think the discussion of hydrologic control on Li isotopes is useful but if find the heavy reliance on the geologic records, in absence of the flux-weighted analysis and a mechanistic discussion, a bit challenging to believe. Residence times, driven by an attendant decrease in discharge, will decrease weathering fluxes so this explanation for Li isotopic shifts also needs to maintain balance in the carbon cycle and it is not clear that it can. In figure 4, for example, the $d^7\text{Li}$ increases in the late Miocene, while weathering fluxes are somewhat invariant. So the key question to me is: How substantial of a shift in hydrology would it take to drive the $d^7\text{Li}$ transitions and why is that not reflected in the weathering flux? I think this question needs to be addressed.

Re: Following Reviewer#3's suggestion, we have added detailed discussion of Li flux, seawater Li residence times, and weathering flux. Reviewer #2 also have helpful explanations on weathering flux (see his comment for lines 196-210). We agree that the weakened hydrology may decrease the continental weathering (Supplementary Fig. 11c). First, we think that the balance in the carbon cycle could be maintained by increase in weathering feedback (through increase reactivity and/or water residence time), solid earth CO_2 degassing (Guo et al., 2021; Sternai et al., 2020) or release of CO_2 through weathering of sedimentary rocks (Horan et al., 2019; Torres et al., 2014). Second, there are studies showing that global weathering flux may have decreased, e.g., marine calcification indicators and the new records in the Himalaya-Tibetan areas show a 50% decrease in the rates of CO_2 consumption during the Neogene (Clift and Jonell, 2021; Si and Rosenthal, 2019). Third, a recent study indicates no need for increased weathering fluxes to reconcile long-term changes of atmospheric CO_2 across the Cenozoic (Komar and Zeebe, 2021). Finally, following the suggestions by Reviewers #2 and #3, we have also added the analysis of *land surface reactivity/ feedback strength*.

These discussions now can clarify and address the weathering flux and carbon balance. (RML: 252-281)

In summary, I appreciated the first discussion and the seasonality in the Li isotope data is interesting in useful, especially if reconciled with the Sr data. I struggled to be convinced by the second argument, about global implications: I think the authors do have an argument about the importance of hydrology for Li, but residence time is likely just one of many factors, and any effects on hydrology have to be considered in light of weathering fluxes and carbon cycle balance. Compared to invoking increased reactivity, that allows for greater sensitivity, the change in discharge invoked here seems harder to reconcile.

Re: After adding the newly measured Sr isotopes (Fig.1), you will find the Sr-Li data can be well reconciled now. We also agree with the importance of the role of surface reactivity, and in the revised manuscript we now emphasise that hydrological change is one of the possible mechanisms (not mutually exclusive with others) that can explain weathering flux and carbon cycle balance. Reviewer#2 also provided helpful interpretation for the land surface reactivity. Please see the reply for Sr data and the weathering flux comments above.

A few detailed comments:

There are a lot of very minor grammatical or typographical corrections that should be made but I assume they would be addressed at a later stage and did not mark them below.

Re: We have checked and corrected the mistakes all through our revision. Thanks for the careful review.

Abstract: Line 22 – awkward phrasing - clarify what “composition” is being referred to.

Re: Corrected. Sorry for the unclear phrasing. It means Li isotopic composition.

Line 84: It is not just residence time, but the distribution of residence times and location of flow paths, e.g., flushing of shallow soils during spring snowmelt are well-known from variations in concentration-discharge relationships.

Re: Agree. The flow paths would change, so we didn't rule it out in the text. We think changes in flow paths will change the path lengths and flow rates, which will in turn change the water-rock contact time. We have added new references related to the description of the residence time. (RML: 85)

Figure 1: I was a little confused by the presentation of sites – can the authors clarify this or maybe consider alternative presentation?

Re: We have reorganized the legend to clarify it in the figure (New Fig.2), and modified the figure by adding notes “upstream to downstream” for the sites for time-series and spatial data.

Line 93: 20% of freshwater but what fraction of Li? Generally, rivers in the tropics are ~80% of global discharge, and presumably global Li flux?

Re: Yes, we have double checked the data. The annual discharges from Amazon (6900 km³) and Congo (1230 km³) rivers account for 22% of global discharge (37000 km³). Their Li flux accounts for 8.4% of global flux.

Line 102: The fact that the Sr shifts indicates different water sourcing within a yearly hydrograph – evidence not simply a straightforward residence time the authors imply?

Re: Yes, we agree with that Sr indicates different water sources. Please see the replies for the Sr comment above, and the lithology effects on new time-series ⁸⁷Sr/⁸⁷Sr for the BH and SL rivers in Fig. 1d.

Line 106: I haven't read all these papers on residence time for Li, but I would guess many of them look at 1-D systems not 3-D watersheds? The authors at line 106 need to clarify mixing – a hydrograph for a river is itself a manifestation of mixing – different fractions of groundwater, interflow and surface runoff. Usually, low discharge is predominated by groundwater so leading to line 108, given that most rocks are the same composition, I don't understand this argument at all. You would need to look at Li mass balance.

Re: We totally agree that a hydrograph for a river is a mixing from groundwater, interflow and surface runoff. Sorry for the misleading. The original authors didn't refer to the mixing from surface runoff and groundwater. They interpret the time-series $\delta^7\text{Li}$ variations of the Congo River as the result of a mixing of two major tributary exhibiting contrasting $\delta^7\text{Li}$ values and whose proportions are changing with time, e.g., one tributary has higher proportion of runoff in wet seasons, and the other has higher proportion in dry seasons. However, the authors also indicate that they cannot provide evidence of seasonal runoff variations for the individual tributaries, because the data are not available.

Despite this, it is clear in their paper that the time-series $\delta^7\text{Li}$ values exhibit significant relationship ($r^2=0.84$) with river discharge, suggesting an ignored importance of hydrology on $\delta^7\text{Li}$.

Here, we want to say that if the seasonal $\delta^7\text{Li}$ were determined by tributary mixing, it will be either increase or decrease from dry to wet seasons as Sr isotopes (Supplementary Fig. 9). In fact, when considering all our seasonal $\delta^7\text{Li}$ data, they show a common decrease from dry to wet seasons. In particular, in one of the world's largest river systems (G-B-M), the $\delta^7\text{Li}$ values at the headwaters of the Ganges, at the downstream of major tributaries (the Ganges, Brahmaputra and Meghna), and the G-B-M mouth, all showed consistent lower $\delta^7\text{Li}$ in wet seasons relative to dry seasons. Because of their consistency, we therefore propose that simple mixing can't explain this phenomenon. We have clarified this in our Supplementary Note 1.

Line 116: the authors cite a study for long and short residence times based on dry and wet seasons, which is intuitive, but this particular study looks at 22 Swiss catchments – for larger scale analysis/models, like continental US, I believe the picture is more

complicated? Anyway, I agree with the general interpretation, but it would be nice to backstop it with basic catchment hydrology – generally old groundwater as baseflow vs. interflow vs. surface runoff, where the latter two supplement or dilute the baseflow at high discharge.

Re: Thanks for the agreement. By checking our dataset for the large river systems, such as the Yenisei, G-B-M, Yellow, and the tropical Congo rivers, they show higher $\delta^7\text{Li}$ in dry than wet seasons (Fig.2, Extended Data Fig. 5). In order to further test the observations in larger scales, we compared the spatial Li isotopes (Fig.3), which also show the overall higher $\delta^7\text{Li}$ in dry mid-high lowlands than in the wet tropical lowland rivers. These observations all support a hydrologic dependence of riverine $\delta^7\text{Li}$. According to the suggestions, we also added a reference (Hindshaw et al., 2019) indicating that dry and wet seasons have long and short residence times, respectively.

Figure 2: is this all the data or are some averaged? I some of these are averaged or one sample it seems a little confusing to plot them if the point is the seasonal data. I was a little confused about what was plotted. In addition, is it really fair to draw a line from 3 averages??? Finally, an important point for the latter arguments in the paper: *is it the flux of Li that matters and if one plots $d^7\text{Li}$ vs Li flux, what does that look like? Active mountains and tropical lowlands have very different Li concentrations – might be a nice additional panel.*

Re: (1) Yes, these are spatial data for rivers in lowlands and mountains, not seasonal. For these spot samples, when a sampling site has multiple available measurements by different studies, we calculate an average value, and others are individual measurement. We have added details to clarify it in the supplementary Data 2; (2) We employed 3 averages to explore the overall trend of runoff vs. $\delta^7\text{Li}$, because they together included 12 rivers in the tropical lowlands across the equator (Amazon, Congo and Orinoco rivers), 27 rivers in the lowlands at middle-to-high latitudes (Mackenzie, Lena, Yenisei, Baikal rivers), and 25 rivers from major orogenic belts (the Andes, the Himalayas-Tibetan Plateau, the New Zealand Alps, and the Rocky and the Machenzie Mountains). Following the suggestion, we have deleted the line from the 3 averages, and this won't affect our result that dry lowlands have overall higher river $\delta^7\text{Li}$ than wet, tropical lowlands; (3) We appreciate the Li flux comment, and have added Li flux discussion as suggested, please see the replies for the comments above. Thanks for the constructive suggestions.

I like the paper from intro to about line 143, although I think the Sr issue and hydrology of basins could be better represented, though I appreciate the desire to keep the argument simple.

Re: We appreciate Reviewer#3 for the understanding and interest.

Figure 3. I am not really sure the Hirnantian glaciation is worth including given how different weathering was at that time and how uncertain the records are, and generally it doesn't add much. The Cenozoic and Cretaceous records make sense. For the speleothem, visually it might be easier to highlight the glacial in red to be consistent

with other records, it is confusing they are blue but Li values go up, then delete the Hirnantian record.

Re: After carefully considering the Reviewer's suggestion, we think it is still useful to keep the Hirnantian event: (1) the data are robust: $\delta^7\text{Li}$ values of both bulk carbonate and brachiopods match well. Their consistency could eliminate the Reviewer#1's concern for diagenesis influences on bulk carbonate; (2) In the scarce existing data with timescales older than the Cenozoic, this is a unique published dry event (with increased $\delta^7\text{Li}$), which provides a valuable, contrasting case to two other wet events (OAEs 1a and 2, with decreased $\delta^7\text{Li}$); (3) It nicely takes the story far back in time, though we agree there are uncertainties; (4) We note that new evidence further supports a drying climate during this event (Li et al., 2021), still consistent with an increasing $\delta^7\text{Li}$ corresponding to a drying climate as shown in our dataset; Finally, keeping this record may stimulate researchers working on paleo-climate events to test it, or use it to reconstruct past hydrologic changes in deep timescales, for which the hydrological records are still limited in our knowledge (e.g., Liu et al., 2020).

We appreciate the color suggestions for Fig. 3. Sorry for misleading. Our design is to use the blue marking the dry events (e.g., glacial periods, Hirnantian events), while red marking the wet events (e.g., PETM, OAEs). In fact, although the time scales are quite different, dry events for both glacial speleothem and Hirnantian glaciation show increased $\delta^7\text{Li}$, whereas wet events show decreased $\delta^7\text{Li}$. We have added notes to the figure legend to avoid misleading.

Line 145: I like this discussion – it is an important piece to address the question of how seasonal variations observed in the first part may or may not translate into LI associated with mean climate states. I think this point should be made more strongly at the beginning, especially because the riverine data could strongly reflect one season once you flux weight it. That said, I don't much about the fidelity or current interpretations of the PETM and OAE marine records and hopefully another review can comment.

Re: Thanks for the interest and suggestions. We have reorganized the first part and added new figures and data at the beginning to display how river $\delta^7\text{Li}$ values are associated with hydrology (Fig.1). We have also added both time-series and spatial Li yield data in our manuscript as suggested (Fig.3 and Extended Data Fig. 13). For the paleo-events, after adding the Li flux and seawater Li residence time suggested by Reviewers #1 and #3, we find that change in continental river $\delta^7\text{Li}$ still can result in a consistent change in the ocean. This helps to explain the long-term seawater Li records. Finally, we note that a new accepted paper regarding PETM demonstrating that the dramatically enhanced hydrology during this event is consistent with a rapid $\delta^7\text{Li}$ negative excursion (Pogge von Strandmann et al., 2021, Science Advances). This again supports our conclusion.

Line 182: Again, I worry about this conclusion... residence time doesn't scale perfectly with climate-driven hydrologic changes and during these times there were other variables, such as topography, vegetation, etc. that change residence time. For example,

residence time (assuming the authors mean residence time and not travel time) is defined as volume of water in storage divided by the discharge – during drier periods water tables are also potentially lower so less water in storage means that residence time could potentially be unchanged. Finally, if you change residence time you likely change the weathering flux, concentrations go up but discharge goes down, and thus given that weathering fluxes scale strongly with discharge it is hard to see how this can balance out.

Re: First, as demonstrated in the manuscript, our modern dataset has already included contrasting climate, vegetation, basin sizes and topography, but all showing that residence time linked to hydrology remains a major control on $\delta^7\text{Li}$. Second, after adding the discussion of Li yield as you suggested, it is clear now as discharge decreases, Li yield decreases, although its concentrations are going up (Supplementary Fig. 13). Third, following the suggestion by Reviewer#1, we have added a box model of variable seawater Li residence times in response to river influx. This helps to explain how change in continental runoff would affect river Li flux and seawater $\delta^7\text{Li}$ values (Supplementary Note 2). Finally, we have also added discussions of weathering flux as suggested by both Reviewers#2 and #3. This explains how change in hydrological cycles impacts water residence times, weathering flux and carbon balance. We hope that these interpretations and discussions can satisfy Reviewer#3. Please see the replies to comments on hydrology, Li fluxes and weathering above.

Line 194-202: Can this be broken down a little bit? Hard to follow.

Re: Revised as suggested. Now it is easier to follow, thanks.

Line 204: I think the authors present evidence for a hydroclimate signal in the Li isotopes – residence time is just one of the potential explanations (or part of a more comprehensive one) so personally I think this is perhaps stepping beyond the evidence presented?

Re: After adding the new measurements of the time-series Sr data (Fig.1), Sr-Li isotopes has been well reconciled to clarify residence times of modern rivers. In particular, the analyses and discussions of river Li flux and variable seawater Li residence times can be consistent to explain the long-term marine Li records. Finally, the suggested discussions of land surface reactivity/feedback strength also addressed the weathering and carbon mass balance. We have included detailed discussions in our revision and hope these analyses and discussions can address Reviewer#3's concern. (RML: 195-202, 234-242, 252-281)

References:

- Caves Rugenstein, J. K., Jost, A. B., Lau, K. V. & Maher, K. Cenozoic carbon cycle imbalances and a variable weathering feedback. *Earth and Planetary Science Letters* **450**, 152-163, doi:10.1016/j.epsl.2016.06.035 (2016).
- Caves Rugenstein, J. K., Ibarra, D. E. & von Blanckenburg, F. Neogene cooling driven by land surface reactivity rather than increased weathering fluxes. *Nature* **571**, 99-102, doi:10.1038/s41586-019-1332-y (2019).

- Clift, P. D. & Jonell, T. N. Himalayan-Tibetan Erosion Is Not the Cause of Neogene Global Cooling. *Geophysical Research Letters* **48**, doi:10.1029/2020gl087742 (2021).
- Dellinger, M. *et al.* Riverine Li isotope fractionation in the Amazon River basin controlled by the weathering regimes. *Geochimica et Cosmochimica Acta* **164**, 71-93, doi:10.1016/j.gca.2015.04.042 (2015).
- Fries, D. M. *Temporal variations in continental weathering processes: Insights from Li and Mg isotopes*, University of Southampton, Doctoral Thesis, (2018).
- Guo, Z., Wilson, M., Dingwell, D. B. & Liu, J. India-Asia collision as a driver of atmospheric CO₂ in the Cenozoic. *Nat Commun* **12**, 3891, doi:10.1038/s41467-021-23772-y (2021).
- Henchiri, S., Gaillardet, J., Dellinger, M., Bouchez, J. & Spencer, R. G. M. Riverine dissolved lithium isotopic signatures in low-relief central Africa and their link to weathering regimes. *Geophysical Research Letters* **43**, 4391-4399, doi:10.1002/2016gl067711 (2016).
- Hindshaw, R. S., Teisserenc, R., Le Dantec, T. & Tananaev, N. Seasonal change of geochemical sources and processes in the Yenisei River: A Sr, Mg and Li isotope study. *Geochimica et Cosmochimica Acta* **255**, 222-236, doi:10.1016/j.gca.2019.04.015 (2019).
- Horan, K. *et al.* Carbon dioxide emissions by rock organic carbon oxidation and the net geochemical carbon budget of the Mackenzie River Basin. *American Journal of Science* **319**, 473-499, doi:10.2475/06.2019.02 (2019).
- Jenkyns, H. C., Dickson, A. J., Ruhl, M., Boorn, S. H. J. M. & Heimhofer, U. Basalt-seawater interaction, the Plenian Cold Event, enhanced weathering and geochemical change: deconstructing Oceanic Anoxic Event 2 (Cenomanian–Turonian, Late Cretaceous). *Sedimentology* **64**, 16-43, doi:10.1111/sed.12305 (2017).
- Kalderon-Asael, B. *et al.* A lithium-isotope perspective on the evolution of carbon and silicon cycles. *Nature* **595**, 394-398, doi:10.1038/s41586-021-03612-1 (2021).
- Komar, N. & Zeebe, R. E. Reconciling atmospheric CO₂, weathering, and calcite compensation depth across the Cenozoic. *Science Advances* **7**, doi:10.1126/sciadv.abd4876 (2021).
- Kump, L. R., & Arthur, M. A. in *Global chemical erosion during the Cenozoic: weatherability balances the budget* In: W, Ruddiman (ed.) in *Tectonic Uplift and Climate Change*. 399–426 Plenum Press: New York. (1997)
- Lechler, M., Pogge von Strandmann, P. A. E., Jenkyns, H. C., Prosser, G. & Parente, M. Lithium-isotope evidence for enhanced silicate weathering during OAE 1a (Early Aptian Selli event). *Earth and Planetary Science Letters* **432**, 210-222, doi:10.1016/j.epsl.2015.09.052 (2015).
- Li, G. & West, A. J. Evolution of Cenozoic seawater lithium isotopes: Coupling of global denudation regime and shifting seawater sinks. *Earth and Planetary Science Letters* **401**, 284-293, doi:10.1016/j.epsl.2014.06.011 (2014).
- Li, N. *et al.* Redox changes in the outer Yangtze Sea (South China) through the Hirnantian Glaciation and their implications for the end-Ordovician biocrisis. *Earth-Science Reviews* **212**, doi:10.1016/j.earscirev.2020.103443 (2021).
- Liu, W. *et al.* Onset of permanent Taklimakan Desert linked to the mid-Pleistocene transition. *Geology* **48**, 782-786, doi:10.1130/g47406.1 (2020).
- Pogge von Strandmann, P. A. E. *et al.* Global climate stabilisation by chemical weathering during the Hirnantian glaciation. *Geochemical Perspectives Letters*, **3**, 230-237, doi:10.7185/geochemlet.1726 (2017a).
- Pogge von Strandmann, P. A. E., Jenkyns, H. C. & Woodfine, R. G. Lithium isotope evidence for

- enhanced weathering during Oceanic Anoxic Event 2. *Nature Geoscience* **6**, 668-672, doi:10.1038/ngeo1875 (2013).
- Pogge von Strandmann, P. A. E. *et al.* Lithium isotope evidence for enhanced weathering and erosion during the Palaeocene-Eocene Thermal Maximum. *Science Advances* (2021).
- Pogge von Strandmann, P. A. E. *et al.* Lithium isotopes in speleothems: Temperature-controlled variation in silicate weathering during glacial cycles. *Earth and Planetary Science Letters* **469**, 64-74, doi:10.1016/j.epsl.2017.04.014 (2017b).
- Si, W. & Rosenthal, Y. Reduced continental weathering and marine calcification linked to late Neogene decline in atmospheric CO₂. *Nature Geoscience* **12**, 833-838, doi:10.1038/s41561-019-0450-3 (2019).
- Sternai, P. *et al.* Magmatic Forcing of Cenozoic Climate? *Journal of Geophysical Research: Solid Earth* **125**, e2018JB016460, doi:10.1029/2018jb016460 (2020).
- Torres, M. A., West, A. J. & Li, G. Sulphide oxidation and carbonate dissolution as a source of CO₂ over geological timescales. *Nature* **507**, 346-349, doi:10.1038/nature13030 (2014).

REVIEWER COMMENTS

Reviewer #2 (Remarks to the Author):

Zhang and coauthors provide new high-temporal resolution river Li isotope data from Tibet and the Pamir plateaus and combine this with a global dataset where seasonal Li isotope data is available to infer how hydrology may influence what we observe in the geologic record. Overall I found the mechanism proposed reasonable, increased fluid residence time on continents associated with cooling leading to increased $d7Li$ (and vice versa for warming). I have fully read through the rebuttal arguments and the new manuscript. The authors have done a thorough job in revision, including new supplementary text that caveats assumptions, as well as addressing much of the phrasing issues I and the other reviewers brought up. I have no additional comments.

Reviewer #4 (Remarks to the Author):

I would like first to specify that I am a new reviewer the 3 reviewers of the manuscript that I discovered as well as the three previous reviews and the detailed response of the authors and their revised version.

After my reading of all the documents, I think that the authors are globally addressed most of the points raised by the reviewers. I am a bit less convinced by the arguments written in the answers to reviewer 3 as part of the observations (if not all) may be due contribution from aquifers. The lack of evidence of binary mixing is not sufficient to demonstrate the lack of contribution either from precipitation (e.g. ice melting having seawater $d7Li$ value) or from local aquifers. The authors recognize themselves the variability of $d7Li$ measured in aquifers. At the end, what matters is the ratio of dissolved Li versus secondary phase Li, and this ratio is very likely due not only to runoff, but also to a number of other variables including vegetation type and activity, topography, etc. These variables being obviously not the same at the seasonal timescale for a given watershed and over the Cenozoic .

So, based on the previous reviews, the authors have globally done a good work at answering them and at adapting their manuscript. However, if I had been one of the primary reviewers of this work, I would have asked to perform much more statistical treatments of the data, at various steps. For instance, the authors show differences in $d7Li$ values during dry or wet seasons, but this relies on a very small number of true time series (see e.g. Supplem. fig. 7) compared to what would be necessary to be statistically relevant. What do these local data represent compared to the whole dataset ? what do they represent compared to the annual Li flux to the ocean? In fig. 2, the minimum would be to show n , the number of considered values for each red rectangle. Another important issue is observations from modern glacial environment and permafrost areas: are they really representative of what is going on over the Cenozoic or during warm events? One way to show the data would be to show all published data, not only the one sampled twice or along a time series.

There is another critical - undiscussed - point, the lack of knowledge of river Li due to anthropogenic activities. Choi et al. in 2019, in Nature Communications, demonstrate that even a relatively low Li concentrations in a large river can be already strongly affected by contamination, and this decreases the river $d7Li$ value. I think that it is not possible to ignore this study when dealing with seasonal variations, and when comparing lowland rivers draining large urban areas with high altitude low population density regions. Knowing that the current amount of Li produced worldwide is equivalent of the river Li flu to the ocean is a problem when interpreted modern river data. Only 1% of this Li is recycled, so most of it ends up in the environment. See also the paper from Schlesinger et al GBC 2021.

There are in the literature several studies, which evidence a close relationship between $d7Li$ and precipitation, in particular during the Quaternary period: Bastian et al 2017 (Sci. rep), 2019 (Chem Geol), 2021 (QSR). These studies also show the complexity of the role of precipitation on Li isotopes,

which strongly depends on the initial soil state and on the insolation and Earth planetary position. The impact on clay or water $\delta^{7}\text{Li}$ is not expected to be the same if soils were already largely developed or if soil thickness was initially small. There are also some threshold effects, concerning the intensity of precipitation. It is very surprising that these findings are not discussed in this manuscript, and how these effects observed from past records may agree or not with the authors interpretation .

In the first part of their introduction, the cited references are not appropriate, please review them. For instance, the role of clay formation and dissolution was inferred much before 2015. For the role of dissolution, the citations do not correspond to the experimental studies, which demonstrate its negligible effect. Also, please see most recent studies on biological fractionation of Li isotopes, including the one of the 2nd author, and revise the corresponding sentence in the text.

Finally, concerning the ocean, I would like to say that if the river Li flux to the ocean change significantly, the Li residence time in the ocean will not be significantly changed. This is because the partition coefficient corresponding to Li incorporation into marine clays is a function of the Li concentration in the ocean. Thus, if more Li arrive from rivers, more oceanic Li will be incorporated into marine sediments. How this can be reconciled with short-term variations of carbonate $\delta^{7}\text{Li}$ values?

To conclude, this manuscript defends an original idea, which is a major role of hydrology and precipitation to explain the Cenozoic $\delta^{7}\text{Li}$ record. However, I advise to revise it in order to be more convincing concerning the recent times and to report an updated and more representative picture of the knowledge of the Li isotopic cycle.

Reviewer #5 (Remarks to the Author):

Zhang et al. present new time series measurements of river dissolved load Li isotope ratios from the Tibetan and Pamir plateaus, alongside a global compilation of reported river dissolved load $\delta^{7}\text{Li}$ values and marine $\delta^{7}\text{Li}$ values over the Phanerozoic, to argue for the consistent control of hydrology on river and marine Li isotope archives, from seasonal to 10⁷-year timescales. Their analyses suggest that changes in continental runoff have an outstanding influence on river (and thereby marine) Li isotope compositions, where high runoff corresponds with high continental Li yields and low $\delta^{7}\text{Li}$ values (and vice versa). Importantly, this relationship is informed by observations across a wide range of Earth surface environments, bringing merits to their primary conclusion. I did not review the first draft of the manuscript, but it is evident that the authors went to great lengths to address previous reviewer comments, notably including complementary Sr isotope data for rivers in their study site (and those from previous studies) and calculations of Li residence time in the ocean. These additions and revisions have strengthened their arguments, but I have several comments that the authors should be able to address before it is ready for publication. Once they are addressed, I foresee this work being a meaningful, novel contribution and an important step in disentangling the complex web of processes that influence the carbon cycle on Earth.

Major comments:

a) The role of lithology

Line 102: Within the critical zones of each catchment, is the lithology unimodal? Could changes in $\delta^{7}\text{Li}_{\text{river}}$ values reflect changes in the $\delta^{7}\text{Li}$ value of the rocks/sediments through which water is flowing and not a change in weathering congruency (not dissimilar to the tributary mixing that you refer to in Line 129)? The Sr isotope data (Fig. 1 a, b) seem to partially suggest this as a possibility. It could be helpful to include another supplementary figure where you plot $\delta^{7}\text{Li}_{\text{river}}$ values against $^{87}\text{Sr}/^{86}\text{Sr}$.

b) The potential influence of groundwater of river Li isotope ratios

Lines 137-140: I'm not entirely sure that groundwater flow can be ruled out as a control your

observations in either the Buha or Shaliu catchments, at least in terms of the way you constructed your argument. Stating that groundwater globally has wide-ranging $\delta^{7}\text{Li}$ values (both greater than and less than those of rivers) does not corroborate the lack of groundwater input in the catchments you've studied. However, here's one way to frame your argument: Occam's razor would suggest that despite groundwater having a range of $\delta^{7}\text{Li}$ values, a consistent decrease in $\delta^{7}\text{Li}_{\text{river}}$ values with increasing discharge suggest that groundwater contributions may not be an important driver of these seasonal variations observed globally. Other lines of evidence for the absence of groundwater contribution should also be provided if available.

c) The role of evaporation

Lines 143-147: Some discussion of the role of evaporation should be included here. Recent work (Xu et al., 2021 Geology) show that evaporation can enrich soil waters in ^{7}Li relative to ^{6}Li via enhanced secondary mineral precipitation and that it can affect river Li fluxes. It is plausible that evaporation is not evident among your samples (something you can interrogate by looking at the major/minor element chemistry or O and H isotopes of your water samples; I'm sure you have the former since you report silicate weathering fluxes). If there is evidence for evaporation occurring, then it does not exactly undermine your argument regarding fluid residence time and its influence on Li isotope ratios; increased evaporation would lead regolith (especially soils) to become partially saturated, thus increasing the times it would take for those waters to transit through the weathering zone. However, this finding would suggest that hydroclimate (i.e., precipitation + temperature changes), and not precipitation alone, drive these observed changes in $\delta^{7}\text{Li}_{\text{river}}$ values with discharge.

d) More mechanistic discussion of the influence of discharge on river Li isotope ratios

Lines 142-147: I think that this explanation is a good start in thinking about how seasonal variations in precipitation and runoff yield distinct Li isotope signatures in river water. Although there has been a litany of studies which clearly articulate the primary role of fluid residence time in modulating silicate weathering fluxes (namely studies involving Kate Maher), I think that at least a tacit acknowledgment of other environmental properties associated with wet/dry seasons is warranted (similar comment made below in minor comment for Lines 158-159). Yes, increasing fluid residence time will likely promote secondary mineral formation, but what about the roles of primary mineral dissolution rates, secondary mineral precipitation rates, pCO_2 or pO_2 in regolith, and even temperature? Bohlin and Bickle (2019) EPSL explore the multivariate effects of changing environmental conditions on Li isotope transfer, so it might benefit you to frame your argument within the context of theirs. Also, considering how river Li/Na changes with runoff would be another helpful proxy for understanding the mechanisms of Li transfer. Including a panel of these data in Fig. 3 and a brief discussion here (or elsewhere in the manuscript) would be helpful. A last suggestion, although not totally conclusive, is to compute the saturation state of various minerals (e.g., with PHREEQC or other software) and see how saturation state of secondary minerals compares with Li isotope ratios and Li/Na.

e) Consideration of flux-weighted continental response to hydrologic change

In reading Reviewer #3's suggestions and your responses to them, I can't help but think that this paper should include some Li flux-weighted calculations for relating seasonal $\delta^{7}\text{Li}_{\text{river}}$ variations to marine $\delta^{7}\text{Li}$ variations in the geologic past. Specifically, it would be interesting to calculate the proportion of each river's cumulative annual Li flux generated during the wet and dry seasons (which you could do at least with all samples that you present in Fig. 2) and then consider these proportions as a function of latitudinal zone; it seems like you have other data (i.e., monthly discharge) that could help you make some reasonable assumptions that would go into the calculation. Since you've demonstrated that silicate weathering fluxes correlate with Li yields (I'm inferring this from Fig. 3c and Supplementary Fig. 11c), you could consider global silicate weathering fluxes (such as those from Gaillardet et al., 1999) in ultimately arriving at some likelihood that these seasonal changes in $\delta^{7}\text{Li}_{\text{river}}$ values (as a function of latitude) may be imprinted on marine $\delta^{7}\text{Li}$ records. Li et al., 2022 EPSL recently demonstrated that midlatitude sites along the East and Southeast Asian margin undergo more significant decreases in weathering intensity from Pliocene to present than lower latitude sites, intimating that climate may be a player in weathering changes in the Neogene and showcasing the relevance of thinking about earth surface responses as a function of latitude. You rightly concede that the absolute magnitude of $\delta^{7}\text{Li}$ excursions in marine records would be beyond the scope of the study, but you can more clearly motivate future work through with these Li-weighted flux

calculations.

Minor comments:

Lines 30-31: provide median/mean $\delta^{7}\text{Li}_{\text{dry}}$ - $\delta^{7}\text{Li}_{\text{wet}}$ (and standard error) – or even a Li- and flux-weighted mean $\delta^{7}\text{Li}_{\text{dry}}$ - $\delta^{7}\text{Li}_{\text{wet}}$ of samples in dataset – to complement the range in this sentence, especially because those maximum and minimum values come from only the Columbia watershed

Line 34: change “regulate” to “regulates”

Line 47: cite Kisakurek et al., 2005 EPSL for evidence of Li being primarily found in silicate minerals

Line 48: The reference used here for assertion of the lack of biological fractionation is misplaced. You should cite any/all of these: Clergue et al., 2015 Chemical Geology; Lemarchand et al., 2010 GCA; Pogge von Strandmann et al., 2016 Chemical Geology; Schmitt et al., 2012 Comptes Rendus Geoscience. It should be noted that Li, Liu & Chadwick (2020) GCA argue for the influence of biological activity in the fractionation of Li isotopes, so it might be worth including this caveat here or when discussing weathering in tropical climates (which are noticeably absent from this study, at least those with both dry and wet season measurements)

Lines 56-57: Include citation for 15 ‰ excursions, unless reference 14 applies to both the large excursions over shorter timescales and the shorter excursions. If it applies to both, cite it after “15 ‰” for added clarity

Lines 68: This statement can be strengthened/appended to by stating that nearly the entire range of $\delta^{7}\text{Li}_{\text{river}}$ values can be found in fluids within a single weathering profile, as illustrated by Golla et al., 2021 EPSL

Line 74: Include Golla et al., 2021 EPSL, Bohlin and Bickle, 2019 EPSL, and Wilson et al., 2021 GCA as references

Lines 76-77: I think the statement, as is, is subtly misleading. Your study is primarily concerned with discerning the role of hydrology in present weathering environments with the implication for past global $\delta^{7}\text{Li}_{\text{river}}$ values. I'd consider removing “and past global $\delta^{7}\text{Li}$ variations” or modifying the sentence to be clearer about what your study's foci are.

Lines 94-100: Sometimes you refer to the measured $\delta^{7}\text{Li}_{\text{river}}$ value and other times you're referring to the variation in $\delta^{7}\text{Li}$ values, which can be confusing. When discussing variations, I would include specific notation, such as $\delta^{7}\text{Li}_{\text{dry}}$ - $\delta^{7}\text{Li}_{\text{wet}}$, for clarity.

Lines 125-127: This sentence is a bit awkwardly structured and could be improved. I think you're trying to articulate that the inconsistent relationships between river $^{87}\text{Sr}/^{86}\text{Sr}$ and discharge globally suggest the influence of lithology on the river Sr isotope composition, but it could be read that $\delta^{7}\text{Li}_{\text{river}}$ values are also functions of lithology.

Lines 147-148: I think this statement, if anything, contradicts your previous line of argumentation and it might be best to exclude it; you previously assert that the decrease in fluid residence time is fundamental in the increases of river $\delta^{7}\text{Li}$ values, and then you follow up by saying large increases in $\delta^{7}\text{Li}$ values are seen over short experiment times. Also, I'm not sure using a reference for the experimental weathering of basalts is relevant to your study sites. If what you are trying to articulate (in referencing this experimental study) is that weathering and Li isotopes respond to changes in climate over short timescales, you should cite Wilson et al., 2021 GCA, who show through analyses of speleothems that variations of soil $\delta^{7}\text{Li}$ values are closely related to changes in climate over the course of a year. Including this citation would support your arguments for fluid residence time.

Lines 158-159: I agree with this but it's probably worth mentioning that other factors during dry seasons (e.g., warmer temperatures, higher soil/subsoil pCO₂ due to respiration) may also contribute to secondary mineral formation.

Lines 162-163: Besides those 3 outliers at high erosion rates for mountain rivers (Fig 3b; any hunch for why these fall off that array?), I agree with this. You could better strengthen this assertion by regressing data from each geomorphic setting individually and comparing the slopes to one another. This is worth at least exploring.

Line 204: This is minutiae, but aligning with Reviewer #3's comments, you'll need to be careful with

conflating fluid transit time with residence time. I know you mean to use them interchangeably, but you should rephrase it to “fluid residence time” for purposes of consistency.

Lines 222-223: Worth including/discussing a new study (Li et al., 2021 PNAS) that argues that marine Be isotopes, when linked with a box model for Be cycling, suggest an increase in continental weathering due to increased erosion during the Neogene. Note this doesn't necessarily agree with your proposed mechanisms discussed starting in Line 252, so some lengthier discussion may be needed.

Line 225: Change “infer” to “inferred”

Line 237: “decrease in continental hydrology” seems a bit vague. Perhaps rephrase to “less intense continental hydroclimate” or a “decrease in continental runoff”.

Lines 257, 268: Include Ramos et al., 2020 GSA Today in citation regarding CO₂ release from the solid Earth (in this case, due to metamorphism in continental arcs) over the Phanerozoic, which also predicts a decrease in CO₂ fluxes across the Cenozoic.

Line 258: Include Kemeny et al., 2021 GCA and Bufe et al., 2021 Nature Geoscience as citations here for sedimentary rock weathering as CO₂ sources.

Comments on Figures:

Figure 1c, d: Could be helpful to indicate which samples come from the wet and dry seasons, perhaps by changing the border color of symbols.

Figure 1d: Include published bulk rock $^{87}\text{Sr}/^{86}\text{Sr}$ next to waters for bedrock in these catchments, if those data exist. Or in the text include the rock ranges of bulk rock $^{87}\text{Sr}/^{86}\text{Sr}$ when water Sr isotope data are interpreted.

Figure 3: Include a fourth panel that shows Li/Na vs. runoff. I think this could improve your arguments of high runoff inducing more congruent weathering.

Supplementary Figure 6: Your axis labels need to be swapped.

Dr. Evan J Ramos (Rice University, Brown University)

Below, we have copied all comments from the reviewers (*black, italics*) followed by our responses in blue text. We have revised the text (*red*) in the manuscript accordingly, and provided the revised manuscript with line numbers (RML). We thank the reviewers for the opportunity to greatly improve our manuscript.

Reviewer #2

Zhang and coauthors provide new high-temporal resolution river Li isotope data from Tibet and the Pamir plateaus and combine this with a global dataset where seasonal Li isotope data is available to infer how hydrology may influence what we observe in the geologic record. Overall I found the mechanism proposed reasonable, increased fluid residence time on continents associated with cooling leading to increased $d7Li$ (and vice versa for warming). I have fully read through the rebuttal arguments and the new manuscript. The authors have done a thorough job in revision, including new supplementary text that caveats assumptions, as well as addressing much of the phrasing issues I and the other reviewers brought up. I have no additional comments.

Re: We thank Reviewer#2 for thinking that we have done a thorough job in the revision.

Reviewer #4:

I would like first to specify that I am a new reviewer the 3 reviewers of the manuscript that I discovered as well as the three previous reviews and the detailed response of the authors and their revised version.

After my reading of all the documents, I think that the authors are globally addressed most of the points raised by the reviewers. I am a bit less convinced by the arguments written in the answers to reviewer 3 as part of the observations (if not all) may be due contribution from aquifers. The lack of evidence of binary mixing is not sufficient to demonstrate the lack of contribution either from precipitation (e.g. ice melting having seawater $d7Li$ value) or from local aquifers. The authors recognize themselves the variability of $d7Li$ measured in aquifers. At the end, what matters is the ratio of dissolved Li versus secondary phase Li, and this ratio is very likely due not only to runoff, but also to a number of other variables including vegetation type and activity, topography, etc. These variables being obviously not the same at the seasonal timescale for a given watershed and over the Cenozoic.

So, based on the previous reviews, the authors have globally done a good work at answering them and at adapting their manuscript. However, if I had been one of the primary reviewers of this work, I would have asked to perform much more statistical treatments of the data, at various steps. For instance, the authors show differences in $d7Li$ values during dry or wet seasons, but this relies on a very small number of true time series (see e.g. Supplem. fig. 7) compared to what would be necessary to be statistically relevant. What do these local data represent compared to the whole dataset? what do they represent compared to the annual Li flux to the ocean? In fig. 2, the minimum would be to show n , the number of considered values for each red rectangle. Another important issue is observations from modern glacial environment and

permafrost areas: are they really representative of what is going on over the Cenozoic or during warm events? One way to show the data would be to show all published data, not only the one sampled twice or along a time series.

Re: Thanks for the thoughtful suggestions. We appreciate the opportunity to clarify our paper. Firstly, we agree that other variables may also contribute to river $\delta^7\text{Li}$ variations. In this study, we observed a consistent hydrology control on riverine $\delta^7\text{Li}$ in global scales in both seasonal and large spatial scales. In temporal terms, we compared time-series/seasonal data from the tropical to the Arctic. (i) We find that time-series $\delta^7\text{Li}$ values in both Tibetan rivers (BH and SL) are very sensitive to changes in hydrology: $\delta^7\text{Li}$ decreases as discharge increases, and vice versa (Fig. 1). (ii) We then compiled other available time-series $\delta^7\text{Li}$ data and find that the Yenisei River (the Arctic), the Yellow River (the Loess Plateau), the Strengbach stream (Vosges Mountains) and the Congo River (tropical rainforest) all showed a consistent result (Supplementary Fig. 6). Although these are only 6 sets of time-series datasets, we think that the rivers considered here are globally representative, because they cover contrasting temperature, vegetation types, geomorphic settings and basin sizes. (iii) To further test the hydrological dependence of seasonal $\delta^7\text{Li}$ at larger spatial scales, we analyzed $\delta^7\text{Li}$ from 10 streams for both dry and wet seasons in the Pamir catchment, and checked all available seasonal data from the headwaters of the Ganges, the tributaries and mainstems of the large Ganges-Brahmaputra River system, and streams in the Columbia River (Fig. 2), including unpublished data of major tributaries and mainstems of the world's largest Amazon River system (Fries, 2018). We surprisingly find that these sampling sites show a same result as time-series datasets, that river $\delta^7\text{Li}$ are consistently higher in low discharge periods and lower in high discharge periods. We note that a new study in the Loire River basin also show a consistent seasonal pattern for all mainstem and tributaries (Millot and Négrel, 2021).

In addition to these seasonal datasets, in spatial terms, we designed two scenarios to further understand the role of hydrology in controlling $\delta^7\text{Li}$. (i) In order to examine the hydrology effect under specific geomorphic setting that is similar to proposed early Cenozoic peneplained terrain, we compiled river $\delta^7\text{Li}$ and runoff data of rivers draining flat lowlands (Greenland Shield, Canadian Shield, Siberian Shield, Amazon Shield, Orinoco Shield). We find that rivers across mid to high latitudes (MHL) show overall higher $\delta^7\text{Li}$ coupled with lower runoff than tropical lowland rivers (lower $\delta^7\text{Li}$ and much higher runoff) (Fig. 3). (ii) As the early Cenozoic peneplained terrain and present high-relief terrain are two typical landscape regimes that were previously employed to interpret seawater $\delta^7\text{Li}$ rise since 60 Ma (Misra and Froelich, 2012), we then compiled $\delta^7\text{Li}$ and runoff data of rivers in tectonically active mountains (Andes, Himalaya, Tibetan Plateau, New Zealand, and Rocky and Mackenzie mountains). The comparisons confirm a clear hydrological signal that $\delta^7\text{Li}$ values are lower in active mountains coupled with higher runoff than dry MHL lowlands. It is also interesting to observe that similar runoff coupled with similar $\delta^7\text{Li}$ values between tropical lowlands and active mountains (Fig. 3).

Therefore, when all these pieces are put together, we find that hydrology is a common factor that is tightly linked to changes in continental river $\delta^7\text{Li}$ from the

Equator to the Arctic in both temporal and spatial scales. Moreover, the spatial comparisons also considered the geomorphic shift from the early Cenozoic flat terrains to the present uplifted terrains.

For the control mechanism of Li isotopes in river water, we emphasize a common signal of hydrology. We do not exclude the contributions from other factors, such as topography, vegetation, mixing, soil thickness, etc. Especially for spatial comparisons (Fig. 3), these contributions may be more prominent. Under a similar runoff setting, the contribution of other factors may begin to emerge and obscure the hydrological signal, e.g., topographic change from mountain to plain may change the residence time of water. It should be noted here that the spatial comparison used annual average runoff due to no available instantaneous runoff, which would introduce some uncertainties. We think these are probably the reason why the correlation is not very high for the single landform or climate background (Fig. 3b). Nevertheless, it does not affect our identification of major hydrological signals.

Based on the above considerations, we do not compare all published spatial $\delta^7\text{Li}$ data, because: (i) river $\delta^7\text{Li}$ varies with seasonal runoff, while existing spatial samples are generally one-time sampling in the flood season, and many of them have no available instantaneous runoff data. Therefore, seasonal comparisons better explore the Li isotopic behavior, which provide valuable details for changes in $\delta^7\text{Li}$ with time and varying environmental conditions; (ii) as suggested by both Reviews #4 and #5, we also agree that other factors may contribute to affect river $\delta^7\text{Li}$. The hydrology signal may be obscured if we simply put all data together using one-time sampling data and annual averaged runoff. With these considerations, we thus compared similar geomorphic settings of lowlands between mid-high latitudes and tropical to examine hydrology effects, and different geomorphic settings between lowlands and active mountains to assess the role of tectonics.

We also agree that binary mixing could have contributions to riverine $\delta^7\text{Li}$. Our seasonal datasets, from tributaries and mainstems of large rivers to very small streams, do not suggest it is a major control (Supplementary Note 1), but we do not exclude it. After carefully considering the Reviewer's comments, we reviewed published papers, and find there is also a study based on time-series data that suggests that seasonal $^{87}\text{Sr}/^{86}\text{Sr}$ reflects mixing proportions of different sources, whilst the changes in water residence time accompanying changes in discharge affect $\delta^7\text{Li}$ by altering the balance between primary mineral dissolution and secondary mineral formation (Hindshaw et al., 2019).

Recent studies reported the impact of vegetation activity in the fractionation of Li isotopes. Meanwhile, we note that they also demonstrated that biological controls on Li isotopic compositions in rivers and seawater should be negligible since the plant Li fraction is much lower than silicate fractions (Li et al., 2020; Steinhoefel et al., 2021). The vegetation effects thus still need to be checked in future studies, but we do not consider that it is a first order control.

For the seasonal data, as demonstrated above, there are 6 sets of time-series ($n=189$) and 72 seasonal datasets, which has already included the mainstems and tributaries of several of the world's largest rivers, i.e., the Amazon, Congo, Ganges, Brahmaputra,

Yenisei, Yellow Rivers. Following the suggestion, we have calculated the total annual Li flux of these large rivers (1.44×10^9 mol/yr), which accounts for ~52% of the Li flux for major world rivers estimated by Hu et al. (1998), and ~14% of the annual Li flux to the ocean. We have added discussion in our text. (RML: 121-125)

In terms of the glacial and permafrost data, our interpretation is that it is just one of the considered climate settings across latitudes, and in summer, the temperature in these areas could increase to >15 °C. Besides, the contrasting seasonal climate variations may provide critical clues for in-depth understanding of glacial cycles or climate events.

Finally, we have added the number of considered $\delta^7\text{Li}$ values for each basin in Fig. 2 and the statistics of annual Li flux as suggested. We now have also added clarification that other factors could contribute together to affect river $\delta^7\text{Li}$. (RML: 207-210).

There is another critical - undiscussed - point, the lack of knowledge of river Li due to anthropogenic activities. Choi et al. in 2019, in Nature Communications, demonstrate that even a relatively low Li concentrations in a large river can be already strongly affected by contamination, and this decreases the river $d^7\text{Li}$ value. I think that it is not possible to ignore this study when dealing with seasonal variations, and when comparing lowland rivers draining large urban areas with high altitude low population density regions. Knowing that the current amount of Li produced worldwide is equivalent of the river Li flu to the ocean is a problem when interpreted modern river data. Only 1% of this Li is recycled, so most of it ends up in the environment. See also the paper from Schlesinger et al GBC 2021.

Re: Yes, this is an excellent point that we had ignored. As demonstrated in the reference, study of anthropogenic activities on river Li remains quite limited. According to the suggestions, we have assessed the anthropogenic influences by exploring potential Li pollution in the world's third largest river (in length) Yangtze River. In particular, this river has most dense population and industry in China at the middle to lower reaches (lowland areas). Our preliminary assessment shows that anthropogenic activities do not seem to have a widespread impact on large river basins with high population density.

It is also interesting to note a new case study showing Li pollution in a small catchment within the Loire River basin, but the spatial $\delta^7\text{Li}$ in the entire Loire basin also shows systematically higher values in low flow stages and lower values in high flow stages (Millot and Négrel, 2021), again consistent with our seasonal observations.

We have added the discussion as suggested (supplementary Note 3). Thanks for the constructive comments for assessing the anthropogenic impacts. (RML: 157-160)

There are in the literature several studies, which evidence a close relationship between $d^7\text{Li}$ and precipitation, in particular during the Quaternary period: Bastian et al 2017 (Sci. rep), 2019 (Chem Geol), 2021 (QSR). These studies also show the complexity of the role of precipitation on Li isotopes, which strongly depends on the initial soil state and on the insolation and Earth planetary position. The impact on clay or water $d^7\text{Li}$ is not expected to be the same if soils were already largely developed or if soil thickness was initially small. There are also some threshold effects, concerning the intensity of

precipitation. It is very surprising that these findings are not discussed in this manuscript, and how these effects observed from past records may agree or not with the authors interpretation.

Re: We totally agree with the comments. We have also noted these pioneer papers regarding clay Li isotopic records, including the sedimentary records at the Yangtze Delta and Himalayan basins (Dosseto et al., 2015; Yang et al., 2021). We acknowledge these pioneer studies have launched excellent attempts to decipher the fingerprint of past climate and weathering history. In this study, we didn't discuss these records because in-depth understanding variations history of clay $\delta^7\text{Li}$ in sedimentary records are still uneasy: **i)** unlike large variations of dissolved $\delta^7\text{Li}$, clay $\delta^7\text{Li}$ in sedimentary records generally vary in relative narrow ranges, e.g., Himalaya basin: 3 ‰ to -4 ‰ (Dosseto et al., 2015), Nile basin: 3 ‰ to 1 ‰ (Bastian et al., 2021; Bastian et al., 2017), Yangtze Delta: -2.5 ‰ to -1.1 ‰ (Yang et al., 2021); **ii)** in addition to climate change, $\delta^7\text{Li}$ in clays at different sedimentary settings could also be partly linked to mineral types and/or sources, e.g., negligible changes of clay sources were reported in Nile basin, while in the Yangtze Delta, the $\delta^7\text{Li}$ fluctuations over the last 2 ka are best explained by source changes (Yang et al., 2021) and Clay $\delta^7\text{Li}$ in modern time-series suspended sediments in the Loire river are suggested to reflect changes in mineral phases/source (Millot and Négrel, 2021). These findings suggest that the transport and deposition processes of clays remains complicated. 3) as demonstrated by Yang et al. (2021) that the chronology may bear large uncertainties for sedimentary records. These studies also call for more work to better constrain clay $\delta^7\text{Li}$ imprint in the future.

In this paper we therefore focus on Li isotopes of continental rivers which are sensitive to environmental changes with larger variations. Besides, we also agree with the point about threshold effects of precipitation intensity. We note that for time-series riverine data in individual basins, the $\delta^7\text{Li}$ decreases to overall lower values in rainy season, and then keeps basically stable, even though the rainfall is stronger during this period.

In spite of this, we observe that for both Nile (Bastian et al., 2021) and Himalaya (Dosseto et al., 2015) records, if only considering the major trends of clay $\delta^7\text{Li}$ from 110ka to the LGM, the drying climate is roughly matched by an overall decreasing clay $\delta^7\text{Li}$. This is in line with our interpretation that overall drying climate with longer residence times results in more ^6Li into clays and therefore lower clay $\delta^7\text{Li}$. In any case, we agree that these pioneer studies provide great potential for using clay Li isotopes to trace past climate change.

In the first part of their introduction, the cited references are not appropriate, please review them. For instance, the role of clay formation and dissolution was inferred much before 2015. For the role of dissolution, the citations do not correspond to the experimental studies, which demonstrate its negligible effect. Also, please see most recent studies on biological fractionation of Li isotopes, including the one of the 2nd author, and revise the corresponding sentence in the text.

Re: Thanks for the very careful review. We now have corrected the citations and revised the sentence following the suggestions. (RML: 48 and 55)

Finally, concerning the ocean, I would like to say that if the river Li flux to the ocean change significantly, the Li residence time in the ocean will not be significantly changed. This is because the partition coefficient corresponding to Li incorporation into marine clays is a function of the Li concentration in the ocean. Thus, if more Li arrive from rivers, more oceanic Li will be incorporated into marine sediments. How this can be reconciled with short-term variations of carbonate $\delta^7\text{Li}$ values?

Re: Thanks for the suggestion. We do not rule out this possibility. Following reviewer#1's suggestion "the impact of a variable seawater Li residence time on the interpretation drawn by the authors should be incorporated in the manuscript. We are dealing with a classical scenario of changes in river flux and composition vs. seawater $\delta^7\text{Li}$ composition. The rate of change of $\delta^7\text{Li}$ of seawater is expected to be a direct function of Li residence time.", we therefore added a discussion of variable seawater Li residence time. Based on our estimation (Supplementary Note 2), we have also proposed that during a wet climate, $\delta^7\text{Li}$ from continental rivers to the ocean would decline. Therefore, if the residence time keeps stable, this scenario will cause the decrease of seawater Li isotopes in wet periods. This is consistent with the new published paper that during the PETM, a short period lasting for ~100 ka with dramatically strengthened precipitation and continental runoff, the seawater $\delta^7\text{Li}$ decreased by ~3‰ (Pogge von Strandmann et al., 2021).

To conclude, this manuscript defends an original idea, which is a major role of hydrology and precipitation to explain the Cenozoic $\delta^7\text{Li}$ record. However, I advise to revise it in order to be more convincing concerning the recent times and to report an updated and more representative picture of the knowledge of the Li isotopic cycle.

Re: we appreciate Reviewer #4 for the thoughtful and constructive comments. In our view, all the comments have helped to improve the manuscript considerably by adding clarity to the interpretations, and noting some critical points (e.g., other control factors, more data statistics, human input, etc.) that we had not addressed in the previous version.

Reviewer #5

Zhang et al. present new time series measurements of river dissolved load Li isotope ratios from the Tibetan and Pamir plateaus, alongside a global compilation of reported river dissolved load $\delta^7\text{Li}$ values and marine $\delta^7\text{Li}$ values over the Phanerozoic, to argue for the consistent control of hydrology on river and marine Li isotope archives, from seasonal to 10^7 -year timescales. Their analyses suggest that changes in continental runoff have an outstanding influence on river (and thereby marine) Li isotope compositions, where high runoff corresponds with high continental Li yields and low $\delta^7\text{Li}$ values (and vice versa). Importantly, this relationship is informed by observations across a wide range of Earth surface environments, bringing merits to their primary conclusion. I did not review the first draft of the manuscript, but it is evident that the authors went to great lengths to address previous reviewer comments, notably including complementary Sr isotope data for rivers in their study site (and those from previous studies) and calculations of Li residence time in the ocean. These additions and revisions have strengthened their arguments, but I have several comments that the

authors should be able to address before it is ready for publication. Once they are addressed, I foresee this work being a meaningful, novel contribution and an important step in disentangling the complex web of processes that influence the carbon cycle on Earth.

Re: We thank Reviewer#5 to recommend that our work is a novel contribution and an important step. In particular, the constructive suggestions of mechanistic discussions, mineral saturation, groundwater, evaporation, as well as thoughtful suggestions for many details that helps greatly strengthen our manuscript.

Major comments:

a) The role of lithology

Line 102: *Within the critical zones of each catchment, is the lithology unimodal? Could changes in $\delta^7\text{Li}_{\text{river}}$ values reflect changes in the $\delta^7\text{Li}$ value of the rocks/sediments through which water is flowing and not a change in weathering congruency (not dissimilar to the tributary mixing that you refer to in Line 129)? The Sr isotope data (Fig. 1 a, b) seem to partially suggest this as a possibility. It could be helpful to include another supplementary figure where you plot $\delta^7\text{Li}_{\text{river}}$ values against $^{87}\text{Sr}/^{86}\text{Sr}$.*

Re: The study areas are multi-lithological catchments. We think the $\delta^7\text{Li}_{\text{river}}$ values do not reflect changes in the $\delta^7\text{Li}$ value of rock/sediments here, because primary rock dissolution would not cause isotope fractionation, and the $\delta^7\text{Li}_{\text{river}}$ values are generally higher than the bedrocks/sediments, with the latter showing much more homogeneous values (Dellinger et al., 2015; Murphy et al., 2019; Vigier et al., 2009). We have also added a figure to show the variation between $\delta^7\text{Li}_{\text{river}}$ and $^{87}\text{Sr}/^{86}\text{Sr}$ as suggested (Supplementary Fig. 3), and no correlation is observed. We note that recent new studies further confirmed that Li isotopes cannot be used as lithological tracers of river waters (Choi et al., 2019; Millot and Négrel, 2021). Thanks for the helpful suggestions.

b) The potential influence of groundwater of river Li isotope ratios

Lines 137-140: *I'm not entirely sure that groundwater flow can be ruled out as a control your observations in either the Buha or Shaliu catchments, at least in terms of the way you constructed your argument. Stating that groundwater globally has wide-ranging $\delta^7\text{Li}$ values (both greater than and less than those of rivers) does not corroborate the lack of groundwater input in the catchments you've studied. However, here's one way to frame your argument: Occam's razor would suggest that despite groundwater having a range of $\delta^7\text{Li}$ values, a consistent decrease in $\delta^7\text{Li}_{\text{river}}$ values with increasing discharge suggest that groundwater contributions may not be an important driver of these seasonal variations observed globally. Other lines of evidence for the absence of groundwater contribution should also be provided if available.*

Re: We appreciate the helpful comments and have revised it as suggested. (RML: 151-156).

c) The role of evaporation

Lines 143-147: *Some discussion of the role of evaporation should be included here. Recent work (Xu et al., 2021 Geology) show that evaporation can enrich soil waters in*

^7Li relative to ^6Li via enhanced secondary mineral precipitation and that it can affect river Li fluxes. It is plausible that evaporation is not evident among your samples (something you can interrogate by looking at the major/minor element chemistry or O and H isotopes of your water samples; I'm sure you have the former since you report silicate weathering fluxes). If there is evidence for evaporation occurring, then it does not exactly undermine your argument regarding fluid residence time and its influence on Li isotope ratios; increased evaporation would lead regolith (especially soils) to become partially saturated, thus increasing the times it would take for those waters to transit through the weathering zone. However, this finding would suggest that hydroclimate (i.e., precipitation + temperature changes), and not precipitation alone, drive these observed changes in $\delta^7\text{Li}_{\text{river}}$ values with discharge.

Re: Yes, we appreciate and totally agree with the constructive comments. Following the suggestion, we have directly added monthly averaged evaporation data of the BH and SL catchments during 1990 to 2015 in the revision (Fig. R1). The field data in these two catchments show higher evaporation in wet seasons, which would expect an increased riverine $\delta^7\text{Li}$ in wet season based on the evaporation effect. This is opposite to the consistently lower $\delta^7\text{Li}$ during the period in the BH and SL. Another phenomenon is that the evaporation values in these two catchments are almost the same, but their overall $\delta^7\text{Li}$ values are different and follow a significant relationship between $\delta^7\text{Li}$ and discharge (Fig. 1c). We also noted that in humid regions where evaporation effect is not obvious, e.g., Congo tropical rainforest characterized by an almost stable seasonal temperature, riverine $\delta^7\text{Li}$ remains very sensitive to seasonal Q_w (Supplementary Fig. 11b). Therefore, since evaporation is one of the players of the hydrological cycle, it is not evident for our dataset, consistent with the Reviewer#5's expectation. We have also added the evaporation discussion in the text. Thanks for the thoughtful suggestion that have further improved our understanding of evaporation effect. (RML: 160-168)

Fig. R1 | Monthly evaporation in the BH and SL catchments.

d) More mechanistic discussion of the influence of discharge on river Li isotope ratios

Lines 142-147: I think that this explanation is a good start in thinking about how seasonal variations in precipitation and runoff yield distinct Li isotope signatures in river water. Although there has been a litany of studies which clearly articulate the primary role of fluid residence time in modulating silicate weathering fluxes (namely studies involving Kate Maher), I think that at least a tacit acknowledgment of other environmental properties associated with wet/dry seasons is warranted (similar comment made below in minor comment for Lines 158-159). Yes, increasing fluid residence time will likely promote secondary mineral formation, but what about the roles of primary mineral dissolution rates, secondary mineral precipitation rates, $p\text{CO}_2$ or $p\text{O}_2$ in regolith, and even temperature? Bohlin and Bickle (2019) EPSL explore the multivariate effects of changing environmental conditions on Li isotope transfer, so it might benefit you to frame your argument within the context of theirs. Also, considering how river Li/Na changes with runoff would be another helpful proxy for understanding the mechanisms of Li transfer. Including a panel of these data in Fig. 3 and a brief discussion here (or elsewhere in the manuscript) would be helpful. A last suggestion, although not totally conclusive, is to compute the saturation state of various minerals (e.g., with PHREEQC or other software) and see how saturation state of secondary minerals compares with Li isotope ratios and Li/Na.

Re: We appreciate Reviewer#5 for inspiring us to think more comprehensively. After carefully considering the comment, we checked the previously published relationships between $\delta^7\text{Li}$ and Li/Na, and found that it is not necessary a good relationship between them, in particular for comparisons between basins in large spatial scales (Manaka et al., 2017; Millot and Négre, 2021; Wanner et al., 2014; Wilson et al., 2021). This implies that Li/Na may be also not necessary correlated with runoff. According to the suggestion, we then further checked the relationship between Li/Na and river discharge by using time-series data of the BH, SL and the Yellow Rivers, and indeed found a no clear relationship (Fig. R2b). Please also see replies to the comments on Fig.3.

Fig. R2 | The relationships of river discharge vs. seasonal $\delta^7\text{Li}$ and Li/Na in the BH, SL and Yellow Rivers.

Fig. R3 | The relationships between seasonal $\delta^7\text{Li}$ and saturation index of minerals in the BH, SL and Yellow Rivers.

Following the suggestion, we have also computed the mineral saturation state, and

found no correlations between $\delta^7\text{Li}$ and evaporites (Fig. R3a-b), but weak positive correlations with clay mineral (illite and kaolinite) saturation index (Fig. R3c-d). These time-series datasets support that decreased runoff with longer residence times also enhances saturation state of clay minerals (Fig. R3e-f), resulting in higher riverine $\delta^7\text{Li}$. We have added this discussion in the text and supplementary Fig. 13. In addition, following the suggestions by both Reviewer #4 and #5, we have also added more mechanistic discussion of human activities, dissolution experiments of residence times, evaporation, vegetation, temperature, other factors, etc. in the context. Thanks for the very helpful suggestions for a more mechanistic discussion.

e) Consideration of flux-weighted continental response to hydrologic change

In reading Reviewer #3's suggestions and your responses to them, I can't help but think that this paper should include some Li flux-weighted calculations for relating seasonal $\delta^7\text{Li}_{\text{river}}$ variations to marine $\delta^7\text{Li}$ variations in the geologic past. Specifically, it would be interesting to calculate the proportion of each river's cumulative annual Li flux generated during the wet and dry seasons (which you could do at least with all samples that you present in Fig. 2) and then consider these proportions as a function of latitudinal zone; it seems like you have other data (i.e., monthly discharge) that could help you make some reasonable assumptions that would go into the calculation. Since you've demonstrated that silicate weathering fluxes correlate with Li yields (I'm inferring this from Fig. 3c and Supplementary Fig. 11c), you could consider global silicate weathering fluxes (such as those from Gaillardet et al., 1999) in ultimately arriving at some likelihood that these seasonal changes in $\delta^7\text{Li}_{\text{river}}$ values (as a function of latitude) may be imprinted on marine $\delta^7\text{Li}$ records. Li et al., 2022 EPSL recently demonstrated that midlatitude sites along the East and Southeast Asian margin undergo more significant decreases in weathering intensity from Pliocene to present than lower latitude sites, intimating that climate may be a player in weathering changes in the Neogene and showcasing the relevance of thinking about earth surface responses as a function of latitude. You rightly concede that the absolute magnitude of $\delta^7\text{Li}$ excursions in marine records would be beyond the scope of the study, but you can more clearly motivate future work through with these Li-weighted flux calculations.

Re: We are grateful for this excellent idea. Frankly, when we were preparing this paper, we also discussed Li-weighted flux calculations for dry and wet season at global scale. Unfortunately, we didn't succeed. The major difficulties are as follows: 1) For time-series $\delta^7\text{Li}$ data, we currently only have incomplete monthly runoff data of the Congo and the Yenisei; 2) For the seasonal data, we have no runoff data for the streams of the Columbia, the Pamirs (Gaizi) and the headwaters of the Ganges, and no available monthly runoff for the mainstream and tributaries of the Ganges and Brahmaputra; 3) As a crucial point, how to divide the time boundary between dry and rainy season is also a very challenging question. For example, in the East Asian monsoon regions, dividing May - September or June - August as rainy season would introduce large uncertainties on the Li flux and $\delta^7\text{Li}$ calculations. In addition, the durations of wet vs. dry seasons can be quite different for river basins in different regions (e.g., the durations of rainy seasons between Asia monsoonal regions and Congo tropical rainforest); 4)

Therefore, although the available seasonal dataset across latitudes support a clear hydrological effect, if we want to calculate reliable global fluxes for wet and dry seasons, we obviously need more dense seasonal data of Li isotopes and instantaneous/monthly runoff to constrain the details.

Based on above considerations, we hope to take this attractive suggestion as a future plan and strive to carry out more seasonal sampling in estuaries of global large rivers. For example, we have already launched a plan to collect seasonal samples at the river mouths of the Yangtze and the Yellow Rivers. In any case, we also see the important value of global seasonal flux assessment, as suggested by the reviewer#5. This is undoubtedly a very important direction for work.

Coincidentally, I discussed the recommended reference (Li et al., 2022) with the first author (Fangliang Li) in last summer when he was preparing the manuscript. The paper discussed the relationship between weathering and temperature since 4 Ma, but it is a pity that they are unable to identify the impact on weathering intensity from precipitation, due to lack of continuous precipitation data. We are currently planning the possibility of reconstructing past hydrological change via Li isotope and other proxies in the future.

Overall, our work shows the great potential of using Li isotopes as a hydrological indicator, and is also closely related to the changes of global weathering flux and the carbon cycle. We hope the above restrictions can be understood by the reviewers.

Minor comments:

Lines 30-31: provide median/mean $\delta^7\text{Li}_{\text{dry}}$ - $\delta^7\text{Li}_{\text{wet}}$ (and standard error) – or even a Li- and flux-weighted mean $\delta^7\text{Li}_{\text{dry}}$ - $\delta^7\text{Li}_{\text{wet}}$ of samples in dataset – to complement the range in this sentence, especially because those maximum and minimum values come from only the Columbia watershed

Re: Added as suggested, thanks. (RML: 31)

Line 34: change “regulate” to “regulates”

Re: Revised, thanks.

Line 47: cite Kisakurek et al., 2005 EPSL for evidence of Li being primarily found in silicate minerals

Re: Cited, thanks for the careful review. (RML: 47)

Line 48: The reference used here for assertion of the lack of biological fractionation is misplaced. You should cite any/all of these: Clergue et al., 2015 Chemical Geology; Lemarchand et al., 2010 GCA; Pogge von Strandmann et al., 2016 Chemical Geology; Schmitt et al., 2012 Comptes Rendus Geoscience. It should be noted that Li, Liu & Chadwick (2020) GCA argue for the influence of biological activity in the fractionation of Li isotopes, so it might be worth including this caveat here or when discussing weathering in tropical climates (which are noticeably absent from this study, at least those with both dry and wet season measurements)

Re: Revised as suggested, thanks for the very careful review. For vegetation

fractionation, we note that the authors also indicated that biological controls on Li isotopic compositions in rivers and seawater should be negligible since the plant Li fraction is much lower than silicate fractions (Li et al., 2020). (RML: 48)

Lines 56-57: Include citation for 15 ‰ excursions, unless reference 14 applies to both the large excursions over shorter timescales and the shorter excursions. If it applies to both, cite it after “15 ‰” for added clarity

Re: Cited as suggested, thanks. (RML: 58)

Lines 68: This statement can be strengthened/appended to by stating that nearly the entire range of $\delta^7\text{Li}_{\text{river}}$ values can be found in fluids within a single weathering profile, as illustrated by Golla et al., 2021 EPSL

Re: Revised. Thanks for the great suggestion. (RML: 70-71)

Line 74: Include Golla et al., 2021 EPSL, Bohlin and Bickle, 2019 EPSL, and Wilson et al., 2021 GCA as references

Re: Cited as suggested, thanks. (RML: 77)

Lines 76-77: I think the statement, as is, is subtly misleading. Your study is primarily concerned with discerning the role of hydrology in present weathering environments with the implication for past global $\delta^7\text{Li}_{\text{river}}$ values. I'd consider removing “and past global $\delta^7\text{Li}$ variations” or modifying the sentence to be clearer about what your study's foci are.

Re: Removed as suggested. Thanks for the reminder. (RML: 79)

Lines 94-100: Sometimes you refer to the measured $\delta^7\text{Li}_{\text{river}}$ value and other times you're referring to the variation in $\delta^7\text{Li}$ values, which can be confusing. When discussing variations, I would include specific notation, such as $\delta^7\text{Li}_{\text{dry}}$ - $\delta^7\text{Li}_{\text{wet}}$, for clarity.

Re: Added as suggested, thanks. (RML: 103)

Lines 125-127: This sentence is a bit awkwardly structured and could be improved. I think you're trying to articulate that the inconsistent relationships between river $^{87}\text{Sr}/^{86}\text{Sr}$ and discharge globally suggest the influence of lithology on the river Sr isotope composition, but it could be read that $\delta^7\text{Li}_{\text{river}}$ values are also functions of lithology.

Re: We have reworded the sentence to avoid a misleading text. Thanks. (RML: 132-138)

Lines 147-148: I think this statement, if anything, contradicts your previous line of argumentation and it might be best to exclude it; you previously assert that the increase in fluid residence time is fundamental in the increases of river $\delta^7\text{Li}$ values, and then you follow up by saying large increases in $\delta^7\text{Li}$ values are seen over short experiment times. Also, I'm not sure using a reference for the experimental weathering of basalts is

relevant to your study sites. If what you are trying to articulate (in referencing this experimental study) is that weathering and Li isotopes respond to changes in climate over short timescales, you should cite Wilson et al., 2021 GCA, who show through analyses of speleothems that variations of soil $\delta^7\text{Li}$ values are closely related to changes in climate over the course of a year. Including this citation would support your arguments for fluid residence time.

Re: Sorry for the misleading text. We intended to demonstrate that the experiments provided strong evidence that change in fluid residence time can produce large Li isotopic fractionation in a very short period (days to months). Another experiment of loess dissolution also shows a similar result (Supplementary Fig. 14). This is well consistent with the seasonal time scales. We have rephrased the sentence and also cited the reference. Thanks for the suggestions. (RML: 176-183)

Lines 158-159: I agree with this but it's probably worth mentioning that other factors during dry seasons (e.g., warmer temperatures, higher soil/subsoil $p\text{CO}_2$ due to respiration) may also contribute to secondary mineral formation.

Re: Sorry, here MHL lowlands belong to cold climate settings. We have revised the sentence to avoid misunderstanding. We have added other factors as suggested, e.g., enhanced soil water saturation. (RML: 193-194)

Lines 162-163: Besides those 3 outliers at high erosion rates for mountain rivers (Fig 3b; any hunch for why these fall off that array?), I agree with this. You could better strengthen this assertion by regressing data from each geomorphic setting individually and comparing the slopes to one another. This is worth at least exploring.

Re: Thanks for the thoughtful comment. According to the suggestion, we analyzed the relationship of the 3 outliers, and surprisingly find a significant relationship between $\delta^7\text{Li}$ and runoff in the New Zealand rivers (Fig. R4a). When considering Li isotopic variations between different basins, we agree that other factors as suggested by both Reviewers #4 and #5, e.g., topography, vegetation, mixing, soil thickness, soil $p\text{CO}_2$ etc., may also contribute to the $\delta^7\text{Li}$ variations in river water. However, for individual basin in the New Zealand, significant runoff control still exists.

Based on these observations, if considering the time evolution for individual basin, the $\delta^7\text{Li}$ would change with long-term varying climate. Meanwhile the topography, vegetation and other factors may also vary with time. But if comparing the dry MHL lowlands with the active mountains (a flat vs uplifted geomorphic settings), the overall trends are still dependent on a hydrology change (Fig. R4a). Besides, similar runoffs between flat tropical lowlands and tectonically active mountains have similar $\delta^7\text{Li}$. We therefore suggest if both climate and topography are changing with time, hydrological change remains a major role for river $\delta^7\text{Li}$. This is further strongly supported by the relationship between precipitation and seawater Li records over the Cenozoic (Fig. 5 d and e). We have added the figure as suggested (Supplementary Fig. 16).

Fig. R4 | The relationships between $\delta^7\text{Li}$ and runoff. a Lowlands and active mountain areas. b New Zealand.

Line 204: This is minutiae, but aligning with Reviewer #3's comments, you'll need to be careful with conflating fluid transit time with residence time. I know you mean to use them interchangeably, but you should rephrase it to "fluid residence time" for purposes of consistency.

Re: Revised as suggested, thanks. (RML: 243)

Lines 222-223: Worth including/discussing a new study (Li et al., 2021 PNAS) that argues that marine Be isotopes, when linked with a box model for Be cycling, suggest an increase in continental weathering due to increased erosion during the Neogene. Note this doesn't necessarily agree with your proposed mechanisms discussed starting in Line 252, so some lengthier discussion may be needed.

Re: Revised as suggested, thanks. (RML: 262)

Line 225: Change "infer" to "infers"

Re: Revised. Thanks.

Line 237: "decrease in continental hydrology" seems a bit vague. Perhaps rephrase to

“less intense continental hydroclimate” or a “decrease in continental runoff”.

Re: Revised as suggested, thanks. (RML: 276)

Lines 257, 268: Include Ramos et al., 2020 GSA Today in citation regarding CO₂ release from the solid Earth (in this case, due to metamorphism in continental arcs) over the Phanerozoic, which also predicts a decrease in CO₂ fluxes across the Cenozoic.

Re: Added, thanks. (RML: 296-297)

Line 258: Include Kemeny et al., 2021 GCA and Bufe et al., 2021 Nature Geoscience as a citations here for sedimentary rock weathering as CO₂ sources.

Re: Cited as suggested, thanks. (RML: 297)

Comments on Figures:

Figure 1c, d: Could be helpful to indicate which samples come from the wet and dry seasons, perhaps by changing the border color of symbols.

Re: Revised as suggested, thanks.

Figure 1d: Include published bulk rock ⁸⁷Sr/⁸⁶Sr next to waters for bedrock in these catchments, if those data exist. Or in the text include the rock ranges of bulk rock ⁸⁷Sr/⁸⁶Sr when water Sr isotope data are interpreted.

Re: We have added the ranges in the Supplementary Fig. 3 as suggested, thanks.

Figure 3: Include a fourth panel that shows Li/Na vs. runoff. I think this could improve your arguments of high runoff inducing more congruent weathering.

Re: We appreciate the reviewer’s suggestion. A fourth panel has been included as suggested below (Fig. R5b). We think this panel maybe not necessary, because different minerals could have different Li/Na ratios, so the mineral weathering may release different Li/Na. The main target for this figure is that after observing amazing consistent seasonal Li variations in global rivers (i.e., Amazon (Fries, 2018), Congo, Ganges, Brahmaputra, Yenisei, Yellow, Columbia, Tibetan, Pamirs, Strengbach rivers) from the Arctic to the equator, we intend to check whether large spatial scale also exists for this phenomenon. The panel (a) show dry Mid-High latitude (MHL) lowlands have overall distinct $\delta^7\text{Li}$ values relative to humid tropical lowlands. A further test by runoff confirmed our expectation that dry MHL lowlands with lower runoff have higher $\delta^7\text{Li}$, while tropical lowland rivers with higher runoff have overall lower $\delta^7\text{Li}$ (Panel c).

We do not expect a very high correlation in panel (c), because we also agree with both reviewers #4 and #5 that many other factors, e.g., topography, vegetation, mixing, soil thickness, soil $p\text{CO}_2$ etc, may contribute to affect riverine $\delta^7\text{Li}$. Especially for spatial comparison, these contributions may be more prominent. If the runoffs of two basins are similar, the contribution from other factors may emerge and obscure the hydrological signal, e.g., topographic change from mountain to plain may change the residence time of water. It should be noted here that the spatial comparison used annual average runoff due to lack of instantaneous runoff, which could introduce additional uncertainties between $\delta^7\text{Li}$ and runoff.

In spite of this, the observations between the dry and tropical lowlands are clear that dry climate displayed overall higher $\delta^7\text{Li}$ values. In this consideration, we think that the data supports a major control of runoff in large spatial scale, but do not exclude the influences from other factors. We have added discussions of other factors including adding a Supplementary Fig. 16 for Figure 3 in our text as suggested in the context. We hope that the reviewer agrees with this summary.

Fig. R5 | Riverine $\delta^7\text{Li}$ and Li yield from various geological settings.

Supplementary Figure 6: Your axis labels need to be swapped.

Re: Corrected. Thanks for the careful review.

References:

- Bastian, L., Mologni, C., Vigier, N., Bayon, G., Lamb, H., Bosch, D., Kerros, M.-E., Colin, C., Revel, M., 2021. Co-variations of climate and silicate weathering in the Nile Basin during the Late Pleistocene. *Quaternary Science Reviews* 264.
- Bastian, L., Revel, M., Bayon, G., Dufour, A., Vigier, N., 2017. Abrupt response of chemical weathering to Late Quaternary hydroclimate changes in northeast Africa. *Sci Rep* 7, 44231.
- Choi, H.B., Ryu, J.S., Shin, W.J., Vigier, N., 2019. The impact of anthropogenic inputs on lithium content in river and tap water. *Nat Commun* 10, 5371.
- Dellinger, M., Gaillardet, J., Bouchez, J., Calmels, D., Louvat, P., Dosseto, A., Gorge, C., Alanoca, L., Maurice, L., 2015. Riverine Li isotope fractionation in the Amazon River basin controlled by the weathering regimes. *Geochimica et Cosmochimica Acta* 164, 71-93.
- Dosseto, A., Vigier, N., Joannes-Boyau, R., Moffat, I., Singh, T., Srivastava, P., 2015. Rapid response of silicate weathering rates to climate change in the Himalaya. *Geochemical Perspectives Letters*, 10-19.
- Fries, D.M., 2018. Temporal variations in continental weathering processes: Insights from Li and Mg isotopes, Doctoral dissertation, University of Southampton. University of Southampton, p. 184pp.
- Hindshaw, R.S., Teisserenc, R., Le Dantec, T., Tananaev, N., 2019. Seasonal change of geochemical sources and processes in the Yenisei River: A Sr, Mg and Li isotope study. *Geochimica et Cosmochimica Acta* 255, 222-236.
- Li, F., Yang, S., Breecker, D.O., Ramos, E.J., Huang, X., Duan, Z., Guo, Y., Li, C., Mei, X., 2022. Responses of silicate weathering intensity to the Pliocene-Quaternary cooling in East and Southeast Asia. *Earth and Planetary Science Letters* 578.
- Li, W., Liu, X.-M., Chadwick, O.A., 2020. Lithium isotope behavior in Hawaiian regoliths: Soil-atmosphere-biosphere exchanges. *Geochimica et Cosmochimica Acta* 285, 175-192.
- Manaka, T., Araoka, D., Yoshimura, T., Hossain, H.M.Z., Nishio, Y., Suzuki, A., Kawahata, H., 2017. Downstream and seasonal changes of lithium isotope ratios in the Ganges-Brahmaputra river system. *Geochemistry, Geophysics, Geosystems* 18, 3003-3015.
- Millot, R., Négrel, P., 2021. Lithium isotopes in the Loire River Basin (France): Hydrogeochemical characterizations at two complementary scales. *Applied Geochemistry* 125.
- Misra, S., Froelich, P.N., 2012. Lithium isotope history of Cenozoic seawater: changes in silicate weathering and reverse weathering. *Science* 335, 818-823.
- Murphy, M.J., Porcelli, D., Pogge von Strandmann, P.A.E., Hirst, C.A., Kutscher, L., Katchinoff, J.A., Mörth, C.-M., Maximov, T., Andersson, P.S., 2019. Tracing silicate weathering processes in the permafrost-dominated Lena River watershed using lithium isotopes. *Geochimica et Cosmochimica Acta* 245, 154-171.
- Pogge von Strandmann, P.A.E., Jones, M.T., West, A.J., Murphy, M.J., Stokke, E.W., Tarbuck, G., Wilson, D.J., Pearce, C.R., Schmidt, D.N., 2021. Lithium isotope evidence for enhanced weathering and erosion during the Palaeocene-Eocene

Thermal Maximum. *Science Advances*.

Steinhefel, G., Brantley, S.L., Fantle, M.S., 2021. Lithium isotopic fractionation during weathering and erosion of shale. *Geochimica Et Cosmochimica Acta* 295, 155-177.

Vigier, N., Gislason, S.R., Burton, K.W., Millot, R., Mokadem, F., 2009. The relationship between riverine lithium isotope composition and silicate weathering rates in Iceland. *Earth and Planetary Science Letters* 287, 434-441.

Wanner, C., Sonnenthal, E.L., Liu, X.-M., 2014. Seawater $\delta^7\text{Li}$: A direct proxy for global CO₂ consumption by continental silicate weathering? *Chemical Geology* 381, 154-167.

Wilson, D.J., Pogge von Strandmann, P.A.E., White, J., Tarbuck, G., Marca, A.D., Atkinson, T.C., Hopley, P.J., 2021. Seasonal variability in silicate weathering signatures recorded by Li isotopes in cave drip-waters. *Geochimica et Cosmochimica Acta*.

Yang, C., Vigier, N., Yang, S., Revel, M., Bi, L., 2021. Clay Li and Nd isotopes response to hydroclimate changes in the Changjiang (Yangtze) basin over the past 14,000 years. *Earth and Planetary Science Letters* 561.

REVIEWER COMMENTS

Reviewer #5 (Remarks to the Author):

Zhang et al. present time series measurements of river dissolved load Li isotope ratios from the Tibetan and Pamir plateaus, along with a global compilation of reported river dissolved load $\delta^{7}\text{Li}$ values and marine $\delta^{7}\text{Li}$ values over the Phanerozoic, to show that continental hydrology has a direct and consistent effect on river and marine Li isotope archives, from seasonal to million-year timescales. Specifically, they demonstrate that changes in continental runoff have an outstanding influence on river (and thereby marine) Li isotope compositions, where high runoff corresponds with high continental Li yields and low $\delta^{7}\text{Li}$ values (and vice versa). I reviewed the second draft that these authors had submitted and upon reading their responses to my and other reviewers' comments and the updated manuscript, I can conclude that they did a thorough and sufficient job with addressing our suggestions. Their discussion involves a more mechanistic discussion of the drivers of these Li isotope trends (both with determinations of mineral saturation and exploration of evaporation), their concession of not being able to determine weighted global river Li fluxes was well argued, and their supplementary material more clearly rules out other drivers of Li isotope trends. I have no additional comments and I look forward to seeing this in print.

Below, we have copied the final comments from the reviewers (*black, italics*) followed by *our responses in blue text*. We thank the reviewers for the opportunity to greatly improve our manuscript.

Reviewer #5 (Remarks to the Author):

Zhang et al. present time series measurements of river dissolved load Li isotope ratios from the Tibetan and Pamir plateaus, along with a global compilation of reported river dissolved load $\delta^7\text{Li}$ values and marine $\delta^7\text{Li}$ values over the Phanerozoic, to show that continental hydrology has a direct and consistent effect on river and marine Li isotope archives, from seasonal to million-year timescales. Specifically, they demonstrate that changes in continental runoff have an outstanding influence on river (and thereby marine) Li isotope compositions, where high runoff corresponds with high continental Li yields and low $\delta^7\text{Li}$ values (and vice versa). I reviewed the second draft that these authors had submitted and upon reading their responses to my and other reviewers' comments and the updated manuscript, I can conclude that they did a thorough and sufficient job with addressing our suggestions. Their discussion involves a more mechanistic discussion of the drivers of these Li isotope trends (both with determinations of mineral saturation and exploration of evaporation), their concession of not being able to determine weighted global river Li fluxes was well argued, and their supplementary material more clearly rules out other drivers of Li isotope trends. I have no additional comments and I look forward to seeing this in print.

Re: We thank Reviewer#5 for thinking that we have done a thorough job in the revision, and agreeing the publication with no additional comments.